# Optimal Stopping for Sequential Bayesian Experimental Design

## Abstract

In sequential Bayesian experimental design, the number of experiments is usually fixed in advance. In practice, however, campaigns may terminate early, raising the fundamental question: *when should one stop*? Threshold-based rules are simple to implement but inherently myopic, as they trigger termination based on a fixed criterion while ignoring the expected future information gain that additional experiments might provide. We develop a principled Bayesian framework for optimal stopping in sequential experimental design, formulated as a Markov decision process where stopping and design policies are jointly optimized. We prove that the optimal rule is to stop precisely when the immediate terminal reward outweighs the expected continuation value. To learn such policies, we introduce a policy gradient method, but show that naïve joint optimization suffers from circular dependencies that destabilize training. We resolve this with a curriculum learning strategy that gradually transitions from forced continuation to adaptive stopping. Numerical studies on a linear-Gaussian benchmark and a contaminant source detection problem demonstrate that curriculum learning achieves stable convergence and outperforms vanilla methods, particularly in settings with strong sequential dependencies.

## 1 Introduction

Sequential experimentation plays a central role in science and engineering, from clinical trials (Murphy, 2003) and materials discovery (Lookman et al., 2019) to environmental monitoring (Krause et al., 2008). In these settings, practitioners adaptively choose new experiments based on outcomes of earlier ones. A fundamental yet often overlooked question is: *when should experimentation stop*?

The prevailing practice is to fix the number of experiments in advance. However, real-world campaigns frequently face uncertain budgets, limited resources, or diminishing returns that make early termination desirable. Simple threshold-based stopping rules are often used in practice, halting experimentation once a predefined criterion (e.g., posterior variance, remaining budget) falls below a cutoff. While convenient, such rules are inherently myopic: they depend only on the current state and ignore the expected long-term trade-off between the value of additional information and the cost of further experiments. As a result, thresholds can terminate too early, missing valuable insights, or too late, wasting resources (see Figure 1).

A more principled approach is to treat stopping as part of the sequential decision-making process, naturally leading to a *Bayesian optimal stopping formulation* that weighs the immediate reward of terminating against the expected value of continuing. While optimal stopping theory is well developed in stochastic processes (e.g., Peskir & Shiryaev 2006), its integration with Bayesian experimental design remains largely unexplored. Existing sequential design methods typically assumes fixed horizons (see Section 2 for a review), leaving open a fundamental gap: *no general framework jointly optimizes design and stopping in sequential Bayesian experimental design.*

In this work, we close this gap by developing a rigorous framework for optimal stopping in sequential Bayesian experimental design. Our contributions are:

- We cast sequential design with stopping as a Markov decision process (MDP), jointly optimizing design and stopping policies.

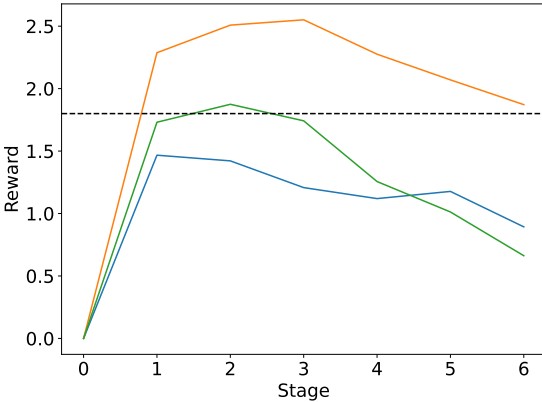

Figure 1: Accumulated reward trajectories over six experiments, illustrating the limitations of threshold-based stopping. Here, a fixed threshold of 1.8 (dashed line) yields very different outcomes across trajectories: the orange path is stopped prematurely after one experiment, missing the higher rewards achievable at stages 2–3; the blue path never crosses the threshold, leading to unnecessary continuation past its peak; the green path happens to cross the threshold near its maximum, but only by chance. Threshold rules are thus highly outcome-dependent, either stopping too early or too late, and fail to account for the expected value of future experiments.

- We prove that the optimal stopping rule is to terminate exactly when the terminal reward exceeds the expected continuation value.
- We introduce an actor-critic policy gradient algorithm for jointly learning design and stopping policies.
- We identify a circular dependency in joint optimization that destabilizes training, and resolve it via curriculum learning that gradually transitions from forced continuation to adaptive stopping.
- Through a linear-Gaussian benchmark and a contaminant source detection problem, we demonstrate that our policy gradient method with curriculum learning achieves stable convergence and outperforms vanilla variants in settings with strong sequential dependencies.

We bridge optimal stopping with Bayesian experimental design, advancing the development of autonomous and resource-efficient experimentation.

## 2 RELATED WORK

Bayesian experimental design (Chaloner & Verdinelli, 1995; Ryan et al., 2016; Alexanderian, 2021; Rainforth et al., 2024; Huan et al., 2024) provides a principled framework for selecting experiments by maximizing expected information gain (Lindley, 1956). Modern sequential extensions (Foster et al., 2021; Ivanova et al., 2021; Blau et al., 2022; Shen & Huan, 2023; Shen et al., 2025) adapt each experiment to previous data, but almost always assume a fixed horizon: the total number of experiments is set in advance, with no explicit stopping policy.

When early termination is considered, it is usually via simple thresholds, such as halting when posterior variance drops below a cutoff or when budget is exhausted. Similar rules appear in active learning (Zhu et al., 2010; Pullar-Strecker et al., 2024), multi-armed bandits (Audibert & Bubeck, 2010), and online A/B testing (Daskalakis & Kawase, 2017). These methods are convenient but *myopic*, ignoring the expected long-term value of further experimentation (Figure 1).

Optimal stopping theory is well studied in stochastic processes and dynamic programming (Peskir & Shiryaev, 2006; Shiryaev, 2008; Haggstrom, 1966; Bertsekas, 2012; Puterman, 2014). Prior reinforcement learning formulations learn stopping policies over a binary action space {continue, stop}, without jointly optimizing the experimental design or control action taken when continuing (Fathan & Delage, 2021; Li & Lee, 2023). Some Bayesian designs include *ad hoc* stopping; for example, Berry et al. (2002) introduced a "terminator" action in adaptive drug trials using low-dimensional

Gaussian approximations and backward induction over small discrete state and action spaces. Their method is specialized and non-information-theoretic. More principled stopping criteria have been developed in the related field of Bayesian Optimization (BO) (Garnett, 2023; Xie et al., 2025). The Knowledge Gradient (KG) (Frazier et al., 2008; Ryzhov et al., 2012) provides a myopic rule that continues sampling only when the expected one-step improvement exceeds the evaluation cost. While useful, KG remains inherently myopic and tied to BO's optimization objective, rather than an information-theoretic or fully optimal stopping formulation like ours.

Despite progress in sequential design and stopping separately, to our knowledge no prior work provides a general framework for jointly optimizing design and stopping in information-theoretic sequential Bayesian experimental design. Our work closes this gap with a principled MDP formulation, an explicit optimal stopping rule, and a stable learning algorithm.

## 3 PROBLEM FORMULATION

### 3.1 PRELIMINARIES

We consider a finite sequence of $N$ experiments indexed by $k = 0, 1, \ldots, N-1$. Let $\theta \in \mathbb{R}^{N_\theta}$ denote model parameters, $\xi_k \in \Xi_k \subseteq \mathbb{R}^{N_\xi}$ the design for experiment $k$, and $y_k \in \mathbb{R}^{N_y}$ the corresponding observation. The information history at stage $k$ is $I_k = \{\xi_0, y_0, \ldots, \xi_{k-1}, y_{k-1}\}$ with $I_0 = \emptyset$. The stage-$k$ belief (the prior for experiment $k$) is $p(\theta|I_k)$. Observations are assumed to follow

$$y_k = G_k(\theta, \xi_k; I_k) + \epsilon_k, \tag{1}$$

where $G_k$ is the forward map (which may depend on $I_k$), and $\epsilon_k$ is an additive observation noise with known density $p_\epsilon$. We assume $\epsilon_k$ is conditionally independent across stages given $(\theta, \xi_k, I_k)$.

Upon observing $y_k$, the belief is updated via Bayes' rule:

$$p(\theta|y_k, \xi_k, I_k) = \frac{p(y_k|\theta, \xi_k, I_k)\, p(\theta|I_k)}{p(y_k|\xi_k, I_k)}, \tag{2}$$

where $p(y_k|\theta, \xi_k, I_k)$ is the likelihood induced by $(G_k, p_\epsilon)$, and $p(y_k|\xi_k, I_k) = \int p(y_k|\theta, \xi_k, I_k)p(\theta|I_k)\,\mathrm{d}\theta$ is the marginal likelihood. The posterior $p(\theta|y_k, \xi_k, I_k)$ then becomes the prior for the next stage, i.e., $p(\theta|I_{k+1})$. This recursive update defines the belief-state dynamics that will underlie our MDP formulation in section 3.2.

### 3.2 MARKOV DECISION PROCESS FOR OPTIMAL STOPPING

We formulate optimal stopping for sequential Bayesian experimental design as an MDP (see Figure 2). This extends existing MDP-based sequential design framework (e.g., Shen & Huan, 2023) by introducing an explicit stopping action that enables early termination of the experimental sequence.

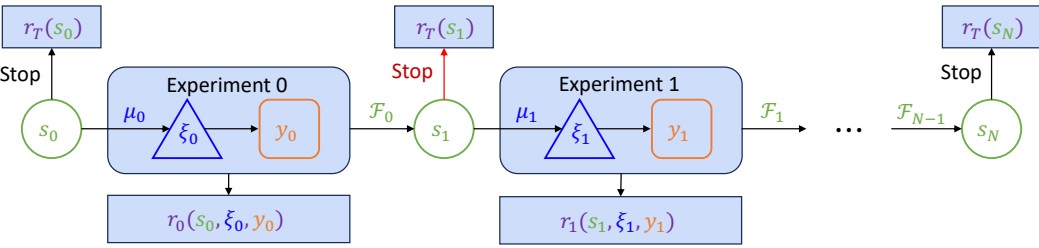

Figure 2: Flowchart of the MDP for sequential Bayesian experimental design with optimal stopping. At each stage, the agent chooses to continue or stop. If continuing, the design policy selects an experiment, yielding an observation and reward, and the state is updated via Bayes' rule. If stopping, a terminal reward is collected. The process repeats until termination or all experiments are exhausted.

**States.** At stage $k$, the state $s_k \in \mathcal{S}$ is represented as $s_k = \{s_k^b, s_k^p\}$. The belief state $s_k^b$ summarizes the uncertainty about $\theta$, which is determined from $I_k$ (for this paper, we can effectively view $s_k = I_k$). The physical state $s_k^p$ captures any additional deterministic variables relevant to design (e.g.,

sensor positions). An additional special terminal state $T$ indicates that the experimental sequence has stopped and no future experiments will be conducted.

**Actions and policies.** At each stage, the agent either (i) terminates the design process, or (ii) chooses a design $\xi_k \in \Xi_k$ to perform an experiment. The stopping policy is a collection of binary decision rules, $\psi = \{\varphi_k : \mathcal{S} \to \{0,1\}, k = 0, \ldots, N-1\}$, where $\varphi_k(s_k) = 1$ indicates stopping at stage $k$. The design policy is a collection of mappings $\pi = \{\mu_k : \mathcal{S} \to \Xi_k, k = 0, \ldots, N-1\}$, which determine designs via $\xi_k = \mu_k(s_k)$.

**State transitions.** When an experiment is performed, the state evolves according to $s_{k+1} = \mathcal{F}_k(s_k, \xi_k, y_k)$, which encodes the Bayesian update in (2) given observation $y_k$. If the termination action is selected, or if all $N$ experiments have been exhausted, the state transitions to the terminal state $T$, where it remains thereafter. Formally,

$$s_{k+1} = \begin{cases} \mathcal{F}_k(s_k, \xi_k, y_k), & s_k \neq T \text{ and } \varphi_k(s_k) = 0, \\ T, & \text{otherwise.} \end{cases} \tag{3}$$

**Rewards.** The reward encodes information gain and experimental cost. Let $r_k(s_k, \xi_k, y_k)$ denote the immediate reward at stage $k$, and $r_T(s_k)$ the terminal reward collected when the experimental sequence ends. Experiment $k$ incurs cost $c_k(\xi_k)$. Information gain is measured using the Kullback–Leibler (KL) divergence between distributions. We consider two equivalent reward formulations:

- *Terminal formulation* (all rewards are accumulated at termination):

$$r_k(s_k, \xi_k, y_k) = 0, \tag{4}$$

$$r_T(s_k) = \begin{cases} D_{\mathrm{KL}}(p_{\theta|I_k} \,\|\, p_{\theta|I_0}) + \sum_{i=0}^{k-1} c_i(\xi_i), & s_k \neq T \text{ and } \varphi_k(s_k) = 1, \\ 0, & \text{otherwise.} \end{cases} \tag{5}$$

- *Incremental formulation* (rewards distributed across stages):

$$r_k(s_k, \xi_k, y_k) = \begin{cases} D_{\mathrm{KL}}(p_{\theta|I_{k+1}} \,\|\, p_{\theta|I_k}) + c_k(\xi_k), & s_k \neq T \text{ and } \varphi_k(s_k) = 0, \\ 0, & \text{otherwise,} \end{cases} \tag{6}$$

$$r_T(s_k) = 0. \tag{7}$$

**Problem statement.** The goal is to find optimal design and stopping policies:

$$\pi^*, \psi^* = \arg\max_{\pi, \psi} U(\pi, \psi) \tag{8}$$

$$\text{subject to} \quad \varphi_k(s_k) \in \{0,1\}, \quad \xi_k = \mu_k(s_k) \in \Xi_k,$$

$$s_{k+1} = \begin{cases} \mathcal{F}_k(s_k, \xi_k, y_k), & s_k \neq T \text{ and } \varphi_k(s_k) = 0, \\ T, & \text{otherwise.} \end{cases}$$

where the objective (expected utility) is the expected total reward:

$$U(\pi, \psi) = \mathbb{E}_{y_{0:N-1}|\pi, \psi, s_0} \left[ \sum_{k=0}^{N-1} (r_k(s_k, \xi_k, y_k) + r_T(s_k)) + r_T(s_N) \right]. \tag{9}$$

Although the terminal and incremental formulations differ in reward structure, they induce equivalent optimal policies when maximizing expected total reward. This MDP formulation thus unifies design and stopping, providing a principled framework for resource-efficient experimentation.

## 4 SOLUTION METHOD

### 4.1 OPTIMAL STOPPING POLICY

Before presenting the optimal stopping rule, we introduce the value function (V-function). The V-function at stage $k$ ($k = 0, \ldots, N-1$) under design policy $\pi$ and stopping policy $\psi$ is:

$$V_k^{\pi, \psi}(s_k) = \mathbb{E}_{y_{k:N-1}|\pi, \psi, s_k} \left[ \sum_{l=k}^{N-1} (r_l(s_l, \mu_l(s_l), y_l) + r_T(s_l)) + r_T(s_N) \right], \tag{10}$$

which represents the expected cumulative reward from state $s_k$ onward when following $\pi$ and $\psi$.

Let us further define $r_T^S(s_k)$ to be $r_T(s_k)$ when $s_k \neq T$ and $\varphi(s_k) = 1$—i.e., the terminal reward obtained by stopping at state $s_k$; and $r_k^C(s_k, \mu_k(s_k), y_k)$ to be $r_k(s_k, \mu_k(s_k), y_k)$ when $s_k \neq T$ and $\varphi(s_k) = 0$—i.e., the immediate reward obtained by continuing.

**Theorem 1** (Optimal Stopping Policy). *For any design policy $\pi = \{\mu_0, \ldots, \mu_{N-1}\}$, the optimal stopping policy is $\psi = \{\varphi_0, \ldots, \varphi_{N-1}\}$ with*

$$\varphi_k(s_k) = \mathbf{1}_{s_k \in \mathcal{T}_k}, \quad k = 0, \ldots, N-1, \tag{11}$$

*where the stopping set $\mathcal{T}_k \subseteq \mathcal{S}$ is defined as:*

$$\mathcal{T}_k = \left\{ s_k \,\middle|\, r_T^S(s_k) \geq \mathbb{E}_{y_k|s_k,\pi} \left[ r_k^C(s_k, \mu_k(s_k), y_k) + V_{k+1}^{\pi,\psi} \big( \mathcal{F}_k(s_k, \mu_k(s_k), y_k) \big) \right] \right\}. \tag{12}$$

We provide a proof in Appendix A.1. Intuitively, the theorem states that the optimal decision at each stage is to compare the terminal reward from stopping with the expected continuation value. Stopping is optimal whenever the former dominates, balancing the value of ending experimentation now against the potential information gain from further experiments.

To formalize this comparison, we define a specialized action-value function (Q-function) that evaluates the return when the next action is to continue—*the value of continuation*—after which $\pi$ and $\psi$ are followed:

$$Q_k^{\pi,\psi}(s_k, \xi_k) = \mathbb{E}_{y_k|s_k,\xi_k} \left[ r_k^C(s_k, \xi_k, y_k) + V_{k+1}^{\pi,\psi}(s_{k+1}) \right], \tag{13}$$

where $s_{k+1} = \mathcal{F}_k(s_k, \xi_k, y_k)$. The stopping set can be expressed equivalently and succinctly as:

$$\mathcal{T}_k = \left\{ s_k \,\middle|\, r_T^S(s_k) \geq Q_k^{\pi,\psi}\big(s_k, \mu_k(s_k)\big) \right\}. \tag{14}$$

Thus the Q-function provides a direct way to evaluate whether stopping and taking the terminal reward is better than the value of continuation.

**Theorem 2** (Terminal-Incremental Equivalence). *Let $U_T(\pi, \psi)$ denote the expected utility under the terminal formulation in (4) and (5), and $U_I(\pi, \psi)$ the expected utility under the incremental formulation in (6) and (7). Then for any policies $(\pi, \psi)$, $U_T(\pi, \psi) = U_I(\pi, \psi)$.*

We provide a proof in Appendix A.2. While terminal-incremental equivalence is known for fixed-horizon sequential design (Foster et al., 2021; Shen & Huan, 2023), extending this result to settings with policy-dependent stopping rules requires proving consistency of optimal stopping across both formulations. This extension is crucial for solving the optimization problem, as it ensures both reward formulations yield equivalent optimal policies and permit flexible implementation choice. We further discuss this tradeoff in section 4.2.

## 4.2 POLICY GRADIENT FOR OPTIMAL STOPPING

Theorem 1 shows that the optimal stopping policy is implicitly determined by the design policy $\pi$. This suggests that the joint optimization problem can be reformulated as a single optimization over the design policy alone. We therefore parameterize the design policy, derive the gradient of the expected utility with respect to its parameters, and employ gradient-based optimization. This leads to a policy gradient (PG) method , which directly optimizes the policy by computing gradients of the objective with respect to the policy parameters (Silver et al., 2014; Wang et al., 2020).

### 4.2.1 POLICY GRADIENT DERIVATION

We parameterize each design function $\mu_k$ with parameters $w_k$, and write $\mu_{k,w_k}$. The full design policy $\pi$ is parameterized by $w = \{w_k\}_{k=0}^{N-1}$, denoted $\pi_w$. The corresponding stopping policy is implicitly determined by $w$ as $\psi_w$, with

$$\varphi_{k,w}(s_k) = \mathbf{1}_{s_k \in \mathcal{T}_{k,w}}, \tag{15}$$

where the stopping set $\mathcal{T}_{k,w}$ is defined by:

$$\mathcal{T}_{k,w} = \left\{ s_k \,\middle|\, r_T^S(s_k) \geq \mathbb{E}_{y_k|\pi_w, s_k}\left[ r_k^C(s_k, \mu_{k,w_k}(s_k), y_k) + V_{k+1}^{\pi_w, \psi_w}\left(\mathcal{F}_k(s_k, \mu_{k,w_k}(s_k), y_k)\right) \right] \right\}$$

$$= \left\{ s_k \,\middle|\, r_T^S(s_k) \geq Q_k^{\pi_w, \psi_w}(s_k, \mu_{k,w_k}(s_k)) \right\}. \tag{16}$$

The expected utility (8) becomes:

$$U(w) = \mathbb{E}_{y_{0:N-1}|\pi_w, \psi_w, s_0}\left[ \sum_{k=0}^{N-1}\left( r_k(s_k, \mu_{k,w_k}(s_k), y_k) + r_T(s_k)\right) + r_T(s_N) \right]. \tag{17}$$

**Theorem 3** (Policy Gradient). *The gradient of the expected utility with respect to the design policy parameters is:*

$$\nabla_w U(w) = \sum_{k=0}^{N-1} \mathbb{E}_{s_k|\pi_w, \psi_w, s_0}\left[ \mathbf{1}_{k<\tau_w} \nabla_w \mu_{k,w_k}(s_k) \nabla_{\xi_k} Q_k^{\pi_w, \psi_w}(s_k, \xi_k)\Big|_{\xi_k = \mu_{k,w_k}(s_k)} \right], \tag{18}$$

*where $\tau_w = \inf\{k : \varphi_{k,w}(s_k) = 1\}$ is the stage when the state first enters the stopping set.*

We provide a proof in Appendix A.3.

### 4.2.2 NUMERICAL ESTIMATION

The gradient in Theorem 3 involves nested expectations over stochastic trajectories and stopping decisions, which are intractable in closed form. We therefore use a Monte Carlo (MC) estimator:

$$\nabla_w U(w) \approx \frac{1}{M} \sum_{m=1}^{M} \sum_{k=0}^{N-1} \mathbf{1}_{k<\tau^{(m)}} \nabla_w \mu_{k,w_k}\left(s_k^{(m)}\right) \nabla_{\xi_k^{(m)}} Q_k^{\pi_w, \psi_w}\left(s_k^{(m)}, \xi_k^{(m)}\right)\Big|_{\xi_k^{(m)} = \mu_{k,w_k}\left(s_k^{(m)}\right)}, \tag{19}$$

where superscript indicates the $m$th sampled trajectory.

Two challenges arise: (1) policy gradients $\nabla_w \mu_{k,w_k}$ must be computed efficiently, and (2) Q-function gradients $\nabla_\xi Q_k^{\pi_w, \psi_w}$ cannot be estimated reliably from nested MC sampling. To address both issues, we adopt a standard actor-critic framework (Sutton & Barto, 2018). In this framework, the "actor" represents the design policy and is optimized via policy gradients (19), while the "critic" learns an approximation of the continuation value (the Q-function) and supplies low-variance gradient information to guide the actor. This structure enables efficient gradient computation without nested simulation and provides a stable mechanism for experimental designs and stopping decisions.

### 4.2.3 ACTOR-CRITIC IMPLEMENTATION

**Policy network.** The design policy is parameterized by a single deep neural network $\mu_w(k, s_k)$. Gradients $\nabla_w \mu_{k,w_k}\left(s_k^{(m)}\right) = \nabla_w \mu_w\left(k, s_k^{(m)}\right)$ are computed via standard backpropagation.

**Q-network.** To approximate the continuation value, we train a neural network $Q_\eta^{\pi_w, \psi_w}(k, s_k, \xi_k)$ with parameters $\eta$. This avoids costly inner MC sampling for directly estimating $Q$ and provides differentiable Q-function estimates. The Q-network is trained by minimizing:

$$\mathcal{L}(\eta) = \frac{1}{M} \sum_{m=1}^{M} \sum_{k=0}^{N-1} \Bigg[ Q_\eta^{\pi_w, \psi_w}\left(k, s_k^{(m)}, \xi_k^{(m)}\right)$$

$$- \left( r_k\left(s_k^{(m)}, \xi_k^{(m)}, y_k^{(m)}\right) + Q_{k+1}^{\pi_w, \psi_w}\left(s_{k+1}^{(m)}, \xi_{k+1}^{(m)}\right) \right) \Bigg]^2 \mathbf{1}_{k<\tau^{(m)}}, \tag{20}$$

with $\xi_k^{(m)} = \mu_w\left(k, s_k^{(m)}\right)$ and $Q_N^{\pi_w, \psi_w}\left(s_N^{(m)}, \cdot\right) = r_T\left(s_N^{(m)}\right)$. This loss enforces the Bellman equation and ensures accurate return predictions.

**Training.** We follow a standard actor-critic loop in Algorithm 1. During training, each episode is generated by first sampling a parameter $\theta^{(m)} \sim p(\theta)$. The design policy $\mu_k(s_k)$ itself is deterministic; however, to encourage exploration during trajectory generation, we inject a small perturbation into the design when generating the Monte Carlo samples used for gradient estimation:

$$\xi_k = \mu_k(s_k) + \epsilon_{\text{explore}}, \quad \epsilon_{\text{explore}} \sim \mathcal{N}(0, \mathbb{I}_{N_\xi} \sigma_{\text{explore}}^2). \tag{21}$$

The value of $\sigma_{\text{explore}}$ reflects the degree of exploration and should be selected based on the problem context. A common practice is to set a large $\sigma_{\text{explore}}$ early in training and reduce it gradually. Stopping is determined when $r_T^S(s_k)$ exceeds the continuation value $Q_k^{\pi,\psi}(s_k, \mu_k(s_k))$ in (14). Policy parameters $w$ are updated via gradient ascent using the MC gradient estimator in (19), while Q-network parameters $\eta$ are updated via stochastic gradient descent on the loss in (20). See Appendix A.4 for more implementation details.

In implementation, the choice between terminal and incremental formulations involves a computational tradeoff. In the terminal-reward formulation, all intermediate rewards are zero, providing sparse supervision to the critic. This can slow convergence early in training, as the Bellman loss (20) receives gradient signal only from episodes that reach the stopping decision. Conversely, the incremental formulation provides denser reward signals at each stage, which can accelerate learning. However, this comes at the cost of computing information gains in $r_k(\cdot)$ at every stage, which is computationally expensive for high-dimensional posteriors and introduce approximation errors in intermediate posterior updates. In practice, the choice depends on the available computational budget and desired learning dynamics. In our experiments, we adopt the terminal formulation for its simplicity and transparent interpretation of the cumulative reward.

Algorithm 1 presents the overall pseudocode of our method.

---

**Algorithm 1** Actor-critic PG for optimal stopping.

1: Set initial state $s_0$, policy updates $L$, MC sample size $M$, policy and Q-network architectures, learning rate $\alpha$ for policy update, exploration scale $\sigma_{\text{explore}}$;
2: Initialize policy and Q-network parameters $w$ and $\eta$;
3: **for** $\ell = 1, \ldots, L$ **do**
4:      Simulate $M$ episodes with path $m = 1, \ldots, M$ following steps 5–10 below;
5:      Initialize $s_0^{(m)} = s_0$ and sample $\theta^{(m)} \sim s_{0,b}$;
6:      **for** $k = 0, \ldots, N-1$ **do**
7:          Sample design $\xi_k^{(m)} = \mu_w(k, s_k^{(m)}) + \epsilon_{\text{explore}}$ and $y_k^{(m)} \sim p(\cdot | \theta^{(m)}, \xi_k^{(m)}, I_k^{(m)})$;
8:          Update state $s_{k+1}^{(m)}$, and compare $r_T^S(s_{k+1}^{(m)})$ with $Q_\eta^{\pi,\psi}(k+1, s_{k+1}^{(m)}, \mu_w(k+1, s_{k+1}^{(m)}))$ for stopping decision;
9:          If $s_{k+1}^{(m)} \in \mathcal{T}_{k+1,w}^{(m)}$ or $k+1 = N$, set $\tau^{(m)} = k+1$ and exit the loop;
10:      **end for**
11:      Store the full information history from all episodes $\{I_{\tau^{(m)}}^{(m)}\}_{m=1}^M$;
12:      Compute and store immediate and terminal rewards for all episodes $\{r_k^{(m)}, r_T^{(m)}\}_{m=1}^M$;
13:      Update $\eta$ by minimizing the loss in (20);
14:      Update $w$ by gradient ascent: $w = w + \alpha \nabla_w U(w)$, where $\nabla_w U(w)$ is estimated via (19);
15: **end for**
16: Return optimized design policy $\pi_w$ and stopping policy $\psi_{w,\eta}$.

---

### 4.3 TRAINING INSTABILITIES AND CURRICULUM LEARNING

Although the PG method provides a principled framework, implementing it in practice reveals training instabilities arising from the problem's inherent circular dependency. The optimal stopping set $\mathcal{T}_k$ in (14) depends on the Q-function $Q_k^{\pi,\psi}(s_k, \mu_k(s_k))$, which itself depends on both the design policy $\pi$ and stopping policy $\psi$. Thus, the stopping policy requires knowledge of the Q-function computed under that very same policy, leading to a fixed-point relationship $\psi^* = f(\pi, \psi^*)$.

In practice, this circularity manifests during joint training of the policy network (parameters $w$) and the Q-network (parameters $\eta$). Early in training, poorly initialized design policies generate weak experiments with little information gain. The Q-network then learns pessimistic continuation values,

which in turn induce premature stopping that prevents policy improvement and entering suboptimal designs. This issue is particularly pronounced in sequential design problems with strong sequential dependencies, where early stopping disrupts the learning of effective long-term strategies.

To overcome this challenge, we adopt a simple form of *curriculum learning*, originally introduced by Bengio et al. (2009), in which an algorithm is gradually exposed to increasingly difficult aspects of the task. We implement this through a stopping probability schedule $p_{\text{stop}}(\ell)$, where $\ell$ indexes training iterations. When the stopping condition $s_k \in \mathcal{T}_k$ is satisfied, the algorithm follows the optimal stopping rule with probability $p_{\text{stop}}(\ell)$, and overrides it (forcing continuation) with probability $(1 - p_{\text{stop}}(\ell))$. This strategy deliberately relaxes stopping early in training, producing longer trajectories that allow both the policy and Q-networks to improve before stopping decisions dominate. As training progresses, $p_{\text{stop}}(\ell)$ is gradually increased, so that stopping behavior converges to the optimal fixed point. In this way, curriculum learning breaks the self-reinforcing cycle of premature stopping and stabilizes training, particularly in settings with strong sequential dependencies.

## 4.4 COMPUTATIONAL CONSIDERATIONS

The policy gradient method requires Monte Carlo trajectory simulation during training to estimate gradients. The primary computational expense comes from evaluating information-theoretic rewards (KL divergences between posteriors) and forward model evaluations. For low-dimensional parameter spaces ($N_\theta \leq 4$), we employ grid-based discretization to compute posteriors and KL divergences, which is both efficient and transparent. For higher-dimensional problems, practitioners can leverage well-established approximation methods such as variational inference (Foster et al., 2019), MCMC-based density estimation, or amortized posterior inference via diffusion models (Baldassari et al., 2023), depending on problem structure and available computational resources.

Once trained, the policy networks enable computationally efficient decision-making. At deployment time, determining the next design and stopping decision requires only a single forward pass through the policy and Q-networks, plus one posterior update and forward model evaluation, eliminating the need for expensive online planning or repeated forward simulations. This efficiency makes the learned policy practical for real-time sequential experimentation, even in resource-constrained settings.

## 5 NUMERICAL EXPERIMENTS

We demonstrate our approach on two numerical examples: a linear-Gaussian benchmark for validation and a sensor movement problem for contaminant source detection that highlights the benefit of curriculum learning. Guidelines for selecting the curriculum learning schedule are provided in Appendix A.6.1.

## 5.1 LINEAR-GAUSSIAN BENCHMARK

We first validate our PG algorithm on a canonical linear-Gaussian benchmark where analytical solutions are available. The forward model is linear in $\theta$ with additive Gaussian noise:

$$y_k = G(\theta, \xi_k) + \epsilon_k = \theta \xi_k + \epsilon_k, \quad \epsilon_k \sim \mathcal{N}(0, 1^2), \tag{22}$$

with prior $\theta \sim \mathcal{N}(0, 3^2)$. Experiments are constrained to designs $\xi_k \in [0.1, 3]$. Conjugacy allows closed-form solutions for optimal policies (see Appendix A.5).

For zero cost ($c_k = 0$), the analytical solution shows that stopping is always optimal at the horizon $N$, since the KL divergence reward is non-decreasing. Figure 3 (left column) shows that our method converges to the maximum stage $N = 3$, achieving the analytical optimum, while naïve threshold policies stop prematurely and underperform.

For negative costs, Figure 3 (middle and right columns) illustrate two cases:

- An $N = 3$ problem with $c_k = -0.5$, where both vanilla and curriculum methods converge to the optimal early stopping at stage 1, consistent with analytical results.

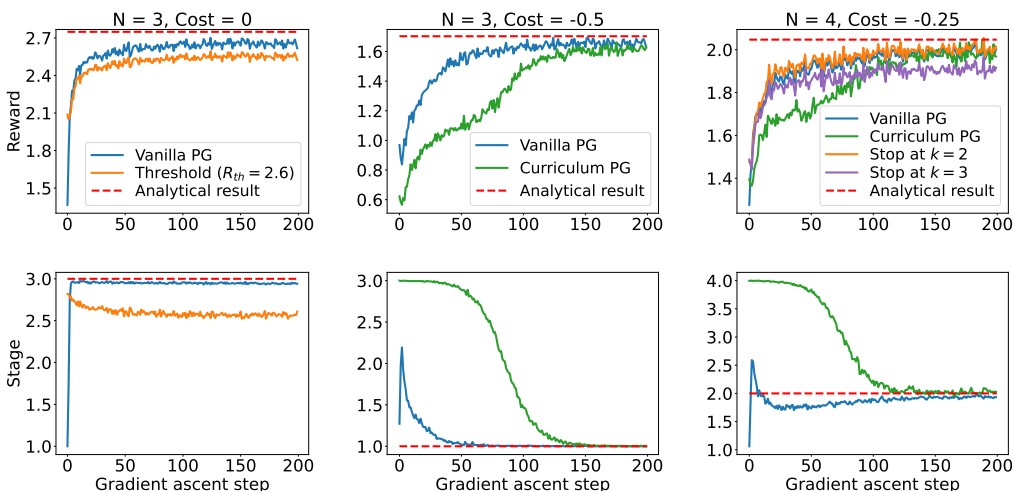

Figure 3: Training convergence of the linear-Gaussian benchmark. Top: average reward; bottom: average stopping stage. Columns correspond to different horizons $N$ and experimental costs.

- An $N = 4$ problem with cost $c_k = -0.25$, where both methods achieve near-optimal performance but with distinct convergence patterns. The vanilla approach often underestimates the optimal stopping stage, becoming trapped below the analytical solution. In contrast, the curriculum method converges more reliably, exploring longer trajectories before settling at the correct stage.

These results illustrate that curriculum learning promotes more stable convergence by mitigating premature stopping during training. The small discrepancy from theory in the $N = 4$ case reflects the flat reward landscape, where stopping at stages 1–3 yields nearly indistinguishable rewards. Even so, the PG method successfully navigates these subtle trade-offs, validating both the algorithm and the role of curriculum learning.

### 5.2 CONTAMINANT SOURCE DETECTION IN CONVECTION-DIFFUSION FIELD

We next consider a sensor-movement problem for contaminant source detection, which exhibits strong sequential dependencies where curriculum learning is essential. Mobile sensors must be strategically repositioned to locate an unknown pollution source, with each placement decision depending critically on previous locations and measurements. Contaminant transport is modeled by a convection-diffusion partial differential equation (PDE) in a two-dimensional square domain with a Gaussian source term (see Appendix A.6.2 for details). We present constant-cost experiments below; design-dependent cost cases are given in Appendix A.6.3.

For zero cost ($c_k = 0$), Figure 4 (left column) shows that both vanilla and curriculum learning converge successfully. The vanilla PG approach quickly settles onto the maximum stage of $N = 4$, while curriculum PG achieves slightly higher rewards. In this regime, continuation rewards dominate termination rewards, even under poor initial designs, so both methods are naturally incentivized to continue and eventually converge.

For negative costs, the training challenges identified in our theoretical analysis become evident. With moderate cost ($c_k = -0.5$, middle column), vanilla PG fails to outperform a fixed stage-4 stopping policy and prematurely convergence to stage 3. In contrast, curriculum learning sustains exploration longer, achieving higher rewards and more appropriate stopping behavior that balances information gain with cost. At higher cost ($c_k = -0.8$, right column), vanilla method degrades further, stopping too early, whereas curriculum learning maintains stable convergence and superior reward.

These results show that sequential dependence and cost-sensitive stopping can destabilize joint optimization, trapping vanilla PG in suboptimal policies. Curriculum learning overcomes this by gradually relaxing stopping constraints, enabling robust convergence from exploration to exploitation.

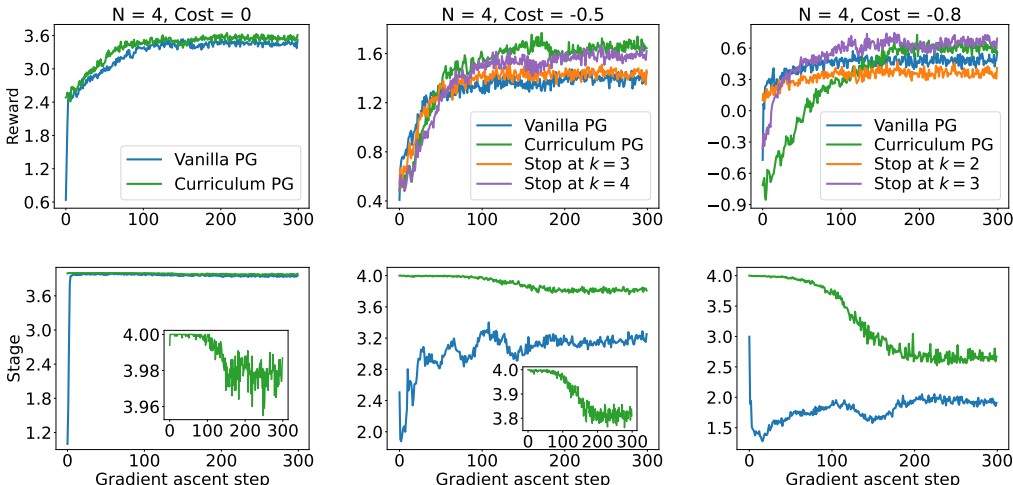

Figure 4: Training convergence of the convection-diffusion source detection problem. Top: average reward; bottom: average stopping stage. Columns correspond to different horizons $N$ and experimental costs.

## 6 CONCLUSION

This work tackles the fundamental challenge of deciding when to terminate sequential experiments in Bayesian experimental design. We developed a principled framework for optimal stopping that accounts for the expected future value of continuation, showing that the optimal rule is simple: continue only when the expected continuation reward exceeds the immediate terminal reward.

From a computational perspective, the coupling between design and stopping policies creates circular dependencies that destabilize training. We addressed this with a PG approach augmented with curriculum learning, which gradually transitions from forced continuation to adaptive stopping. Our experiments demonstrate that, especially for tasks with strong sequential dependencies, curriculum learning achieves stable convergence. The convection-diffusion source detection problem illustrates this setting, where curriculum learning yields substantial gains over fixed stopping rules by breaking the coupling during training.

While our framework offers a principled solution, several limitations point to future work. First, evaluating terminal rewards at every decision stage can be expensive with costly posterior updates or high-dimensional reward structures. Second, alternative formulations that directly parameterize the stopping policy without relying on the optimal stopping theorem (Theorem 1) might avoid circular dependencies and simplify training. Such approaches may converge more slowly or sacrifice theoretical optimality, but could broaden the range of practical applications. Finally, our work focuses on maximizing information gain for parameter inference. However, our framework is highly flexible and could be adapted to goal-oriented experimental designs. For instance, the objective could be modified to specifically improve the prediction of future outcomes, a utility explored by (Kleinegesse & Gutmann, 2021).

Overall, this framework advances autonomous experimental systems by enabling intelligent, resource-aware stopping decisions, improving efficiency of sequential Bayesian experimental design.

### REPRODUCIBILITY STATEMENT

Source code for the implementation of this work, including all algorithms and experimental setups, is available at: https://anonymous.4open.science/r/OS-sOED-E488.

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

# A APPENDIX

CONTENTS

## A.1 PROOF OF THEOREM 1

*Proof of Theorem 1.* For a given design policy $\pi$, consider the V-function under any stopping policy $\psi'$ different from $\psi$:

$$V_N^{\pi,\psi'}(s_N) = r_T(s_N)$$

$$V_k^{\pi,\psi'}(s_k) = \begin{cases} r_T^S(s_k), & \varphi_k'(s_k) = 1, \\ \mathbb{E}_{y_k|\pi,s_k}\left[r_k^C(s_k,\mu_k(s_k),y_k) + V_{k+1}^{\pi,\psi'}(\mathcal{F}_k(s_k,\mu_k(s_k),y_k))\right], & \text{otherwise.} \end{cases} \quad (23)$$

Note that at the terminal stage, $V_N^{\pi,\psi}(s_N) = r_T(s_N) = V_N^{\pi,\psi'}(s_N)$. V-function under the optimal stopping policy $\psi$ satisfies:

$$V_k^{\pi,\psi}(s_k) = \max\left\{r_T^S(s_k), \mathbb{E}_{y_k|\pi,s_k}\left[r_k^C(s_k,\mu_k(s_k),y_k) + V_{k+1}^{\pi,\psi}(\mathcal{F}_k(s_k,\mu_k(s_k),y_k))\right]\right\}. \quad (24)$$

**Base case ($k = N - 1$).** If $\varphi_{N-1}'(s_{N-1}) = 1$, then

$$\begin{aligned} V_{N-1}^{\pi,\psi'}(s_{N-1}) &= r_T^S(s_{N-1}) \\ &\leq \max\left\{r_T^S(s_{N-1}), \mathbb{E}_{y_{N-1}|\pi,s_{N-1}}\left[r_{N-1}^C(s_{N-1},\mu_{N-1}(s_{N-1}),y_{N-1})\right.\right. \\ &\qquad\qquad\qquad\qquad\qquad \left.\left. + V_N^{\pi,\psi}(\mathcal{F}_{N-1}(s_{N-1},\mu_{N-1}(s_{N-1}),y_{N-1}))\right]\right\} \\ &= V_{N-1}^{\pi,\psi}(s_{N-1}). \end{aligned} \quad (25)$$

If $\varphi_{N-1}'(s_{N-1}) = 0$, then

$$\begin{aligned} V_{N-1}^{\pi,\psi'}(s_{N-1}) &= \mathbb{E}_{y_{N-1}|\pi,s_{N-1}}\left[r_{N-1}^C(s_{N-1},\mu_{N-1}(s_{N-1}),y_{N-1})\right. \\ &\qquad\qquad\qquad\qquad \left. + V_N^{\pi,\psi'}(\mathcal{F}_{N-1}(s_{N-1},\mu_{N-1}(s_{N-1}),y_{N-1}))\right] \\ &= \mathbb{E}_{y_{N-1}|\pi,s_{N-1}}\left[r_{N-1}^C(s_{N-1},\mu_{N-1}(s_{N-1}),y_{N-1})\right. \\ &\qquad\qquad\qquad\qquad \left. + V_N^{\pi,\psi}(\mathcal{F}_{N-1}(s_{N-1},\mu_{N-1}(s_{N-1}),y_{N-1}))\right] \\ &\leq \max\left\{r_T^S(s_{N-1}), \mathbb{E}_{y_{N-1}|\pi,s_{N-1}}\left[r_{N-1}^C(s_{N-1},\mu_{N-1}(s_{N-1}),y_{N-1})\right.\right. \\ &\qquad\qquad\qquad\qquad\qquad \left.\left. + V_N^{\pi,\psi}(\mathcal{F}_{N-1}(s_{N-1},\mu_{N-1}(s_{N-1}),y_{N-1}))\right]\right\} \end{aligned}$$

$$= V_{N-1}^{\pi,\psi}(s_{N-1}). \tag{26}$$

Hence, $V_{N-1}^{\pi,\psi}(s_{N-1}) \geq V_{N-1}^{\pi,\psi'}(s_{N-1})$.

**Inductive step.** Suppose $V_{k+1}^{\pi,\psi}(s_{k+1}) \geq V_{k+1}^{\pi,\psi'}(s_{k+1})$. Then

$$V_k^{\pi,\psi}(s_k) = \max\left\{r_T^S(s_k), \mathbb{E}_{y_k|\pi,s_k}\left[r_k^C\left(s_k, \mu_k(s_k), y_k\right) + V_{k+1}^{\pi,\psi}\left(\mathcal{F}_k\left(s_k, \mu_k(s_k), y_k\right)\right)\right]\right\}$$

$$\geq \max\left\{r_T^S(s_k), \mathbb{E}_{y_k|\pi,s_k}\left[r_k^C\left(s_k, \mu_k(s_k), y_k\right) + V_{k+1}^{\pi,\psi'}\left(\mathcal{F}_k\left(s_k, \mu_k(s_k), y_k\right)\right)\right]\right\}$$

$$\geq V_k^{\pi,\psi'}(s_k). \tag{27}$$

By induction, $V_k^{\pi,\psi}(s_k) \geq V_k^{\pi,\psi'}(s_k)$ for all $k = 0, \ldots, N-1$. In particular, $V_0^{\pi,\psi}(s_0) \geq V_0^{\pi,\psi'}(s_0)$, which proves that the stopping policy in Theorem 1 is optimal. $\qquad\square$

## A.2  PROOF OF THEOREM 2

*Proof of Theorem 2.* We first prove the stopping sets are equivalent under two formulations. For convenience, we omit the notation of design policy $\pi$ and stopping policy $\psi$, but distinguish them using upperscript of *incr* and *term* respectively. The policy dependence in the expectations is also omitted for notational consistency. We first prove the following relationship between value functions:

$$V_k^{incr}(s_k) = V_k^{term}(s_k) - \left[D_{\mathrm{KL}}(p_{\theta|I_k}\,||\,p_{\theta|I_0}) + \sum_{i=0}^{k-1} c_i(\xi_i)\right], k = 0, \ldots, N. \tag{28}$$

**Base case ($k = N$).** For the terminal formulation,

$$V_N^{term}(s_N) = r_T(s_N) = D_{\mathrm{KL}}(p_{\theta|I_N}\,||\,p_{\theta|I_0}) + \sum_{i=0}^{N-1} c_i(\xi_i); \tag{29}$$

for the incremental formulation,

$$V_N^{incr}(s_N) = r_T(s_N) = 0. \tag{30}$$

Therefore

$$V_N^{incr}(s_N) = V_N^{term}(s_N) - \left[D_{\mathrm{KL}}(p_{\theta|I_N}\,||\,p_{\theta|I_0}) + \sum_{i=0}^{N-1} c_i(\xi_i)\right]. \tag{31}$$

**Inductive step.** Suppose

$$V_{k+1}^{incr}(s_{k+1}) = V_{k+1}^{term}(s_{k+1}) - \left[D_{\mathrm{KL}}(p_{\theta|I_{k+1}}\,||\,p_{\theta|I_0})) + \sum_{i=0}^{k} c_i(\xi_i)\right]. \tag{32}$$

Then at stage $k$, for the terminal formulation,

$$V_k^{term}(s_k) = \max\left\{D_{\mathrm{KL}}(p_{\theta|I_k}\,||\,p_{\theta|I_0}) + \sum_{i=0}^{k-1} c_i(\xi_i), \mathbb{E}_{y_k}\left[V_{k+1}^{term}(s_{k+1})\right]\right\}; \tag{33}$$

for the incremental formulation,

$$V_k^{incr}(s_k)$$

$$= \max\left\{0, \mathbb{E}_{y_k}\left[D_{\mathrm{KL}}(p_{\theta|I_{k+1}}\,||\,p_{\theta|I_k}) + c_k(\xi_k) + V_{k+1}^{incr}(s_{k+1})\right]\right\}$$

$$= \max\left\{0, \mathbb{E}_{y_k}\left[D_{\mathrm{KL}}(p_{\theta|I_{k+1}}\,||\,p_{\theta|I_k}) + c_k(\xi_k)\right.\right.$$

$$\left.\left. + V_{k+1}^{term}(s_{k+1}) - D_{\mathrm{KL}}(p_{\theta|I_{k+1}}\,||\,p_{\theta|I_0}) - \sum_{i=0}^{k} c_i(\xi_i)\right]\right\}$$

$$= \max \left\{ 0, \ \mathbb{E}_{y_k} \left[ V_{k+1}^{term}(s_{k+1}) - D_{\mathrm{KL}}(p_{\theta|I_k} \,\|\, p_{\theta|I_0}) - \sum_{i=0}^{k-1} c_i(\xi_i) \right] \right\}$$

$$= \max \left\{ D_{\mathrm{KL}}(p_{\theta|I_k} \,\|\, p_{\theta|I_0}) + \sum_{i=0}^{k-1} c_i(\xi_i), \ \mathbb{E}_{y_k} \left[ V_{k+1}^{term}(s_{k+1}) \right] \right\}$$

$$- \left[ D_{\mathrm{KL}}(p_{\theta|I_k} \,\|\, p_{\theta|I_0}) + \sum_{i=0}^{k-1} c_i(\xi_i) \right]$$

$$= V_k^{term}(s_k) - \left[ D_{\mathrm{KL}}(p_{\theta|I_k} \,\|\, p_{\theta|I_0}) + \sum_{i=0}^{k-1} c_i(\xi_i) \right]. \tag{34}$$

By induction, (28) is proved. Using this relationship, we examine the stopping sets. The terminal formulation stopping sets are:

$$\mathcal{T}_k^{term} = \left\{ s_k \ \middle| \ D_{\mathrm{KL}}(p_{\theta|I_k} \,\|\, p_{\theta|I_0}) + \sum_{i=0}^{k-1} c_i(\xi_i) \geq \mathbb{E}_{y_k} \left[ V_{k+1}^{term}(s_{k+1}) \right] \right\}; \tag{35}$$

the incremental stopping sets are:

$$\mathcal{T}_k^{incr} = \left\{ s_k \mid 0 \geq \mathbb{E}_{y_k} \left[ r_k^C(s_k, \xi_k, y_k) + V_{k+1}^{incr}(s_{k+1}) \right] \right\}$$

$$= \left\{ s_k \mid 0 \geq \mathbb{E}_{y_k} \left[ D_{\mathrm{KL}}(p_{\theta|I_{k+1}} \,\|\, p_{\theta|I_k}) + c_k(\xi_k) + V_{k+1}^{incr}(s_{k+1}) \right] \right\}$$

$$= \left\{ s_k \ \middle| \ 0 \geq \mathbb{E}_{y_k} \left[ V_{k+1}^{term}(s_{k+1}) \right] - D_{\mathrm{KL}}(p_{\theta|I_k} \,\|\, p_{\theta|I_0}) - \sum_{i=0}^{k-1} c_i(\xi_i) \right\}$$

$$= \mathcal{T}_k^{term}. \tag{36}$$

Therefore, the stopping sets are equivalent in both formulations. Especially, the optimization objective is

$$U(\pi, \psi) = V_0^{incr}(s_0) = V_0^{term}(s_0), \tag{37}$$

which further proves the equivalence of the optimization problem. $\square$

### A.3 PROOF OF THEOREM 3

To avoid notation congestion, below we will omit the subscript on $w$ and shorten $\mu_{k,w_k}(s_k)$ to $\mu_{k,w}(s_k)$, with the understanding that $w$ takes the same subscript as the $\mu$ function. We also omit explicit conditioning in expectations to save formula space, with the understanding that expectations are taken with respect to the appropriate policies and states.

*Proof of Theorem 3.* The gradient of the expected utility can be written using the V-function:

$$\nabla_w U(w) = \nabla_w V_0^{\pi_w, \psi_w}(s_0). \tag{38}$$

By denoting the V-function as

$$V_k^{\pi_w, \psi_w}(s_k) = \mathbf{1}_{s_k \in \mathcal{T}_{k,w}} r_T^S(s_k)$$
$$+ (1 - \mathbf{1}_{s_k \in \mathcal{T}_{k,w}}) \mathbb{E}_{y_k} \left[ r_k^C(s_k, \mu_{k,w}(s_k), y_k) + V_{k+1}^{\pi_w, \psi_w}(\mathcal{F}_k(s_k, \mu_{k,w}(s_k), y_k)) \right], \tag{39}$$

the gradient of $V_k^{\pi_w, \psi_w}(s_k)$ is

$$\nabla_w V_k^{\pi_w, \psi_w}(s_k)$$

$$= \nabla_w \mathbf{1}_{s_k \in \mathcal{T}_{k,w}} r_T^S(s_k) - \nabla_w \mathbf{1}_{s_k \in \mathcal{T}_{k,w}} \mathbb{E}_{y_k} \left[ r_k^C(s_k, \mu_{k,w}(s_k), y_k) + V_{k+1}^{\pi_w, \psi_w}(\mathcal{F}_k(s_k, \mu_{k,w}(s_k), y_k)) \right]$$

$$+ (1 - \mathbf{1}_{s_k \in \mathcal{T}_{k,w}}) \nabla_w \mathbb{E}_{y_k} \left[ r_k^C(s_k, \mu_{k,w}(s_k), y_k) + V_{k+1}^{\pi_w, \psi_w}(\mathcal{F}_k(s_k, \mu_{k,w}(s_k), y_k)) \right]$$

$$= \nabla_w \mathbf{1}_{s_k \in \mathcal{T}_{k,w}} \left( r_T^S(s_k) - \mathbb{E}_{y_k} \left[ r_k^C(s_k, \mu_{k,w}(s_k), y_k) + V_{k+1}^{\pi_w, \psi_w}(\mathcal{F}_k(s_k, \mu_{k,w}(s_k), y_k)) \right] \right)$$

$$+ \mathbf{1}_{s_k \notin \mathcal{T}_{k,w}} \nabla_w \mathbb{E}_{y_k} \left[ r_k^C(s_k, \mu_{k,w}(s_k), y_k) + V_{k+1}^{\pi_w, \psi_w}(\mathcal{F}_k(s_k, \mu_{k,w}(s_k), y_k)) \right]. \tag{40}$$

The first term in (40) is related to the gradient of an indicator function. To compute this, we define the boundary of the stopping set $\mathcal{T}_{k,w}$ as

$$h(s_k, w) = r_T^S(s_k) - \mathbb{E}_{y_k} \left[ r_k^C(s_k, \mu_{k,w}(s_k), y_k) + V_{k+1}^{\pi_w, \psi_w}(\mathcal{F}_k(s_k, \mu_{k,w}(s_k), y_k)) \right] = 0. \tag{41}$$

The indicator function changes value only on this boundary. Therefore, the gradient can be expressed as a Dirac delta function centered on the boundary:

$$\nabla_w \mathbf{1}_{s_k \in \mathcal{T}_{k,w}} = \delta(h(s_k, w)) \nabla_w h(s_k, w). \tag{42}$$

The first term in (40) is then reduced to $h(s_k, w)\delta(h(s_k, w))\nabla_w h(s_k, w)$, which is always 0 with the property of the Dirac Delta function $x\delta(x) \equiv 0$. For the second term in (40),

$$\nabla_w \mathbb{E}_{y_k} \left[ r_k^C(s_k, \mu_{k,w}(s_k), y_k) + V_{k+1}^{\pi_w, \psi_w}(\mathcal{F}_k(s_k, \mu_{k,w}(s_k), y_k)) \right]$$

$$= \nabla_w \left[ \int_{y_k} p(y_k|s_k, \mu_{k,w}(s_k)) r_k^C(s_k, \mu_{k,w}(s_k), y_k) \, dy_k \right.$$

$$\left. + \int_{s_{k+1}} p(s_{k+1}|s_k, \mu_{k,w}(s_k)) V_{k+1}^{\pi_w, \psi_w}(s_{k+1}) \, ds_{k+1} \right]$$

$$= \nabla_w \int_{y_k} p(y_k|s_k, \mu_{k,w}(s_k)) r_k^C(s_k, \mu_{k,w}(s_k), y_k) \, dy_k$$

$$+ \nabla_w \int_{s_{k+1}} p(s_{k+1}|s_k, \mu_{k,w}(s_k)) V_{k+1}^{\pi_w, \psi_w}(s_{k+1}) \, ds_{k+1}$$

$$= \int_{y_k} \nabla_w \mu_{k,w}(s_k) \nabla_{\xi_k} \left[ p(y_k|s_k, \xi_k) r_k^C(s_k, \xi_k, y_k) \right] \Big|_{\xi_k = \mu_{k,w}(s_k)} dy_k$$

$$+ \int_{s_{k+1}} \left[ p(s_{k+1}|s_k, \mu_{k,w}(s_k)) \nabla_w V_{k+1}^{\pi_w, \psi_w}(s_{k+1}) \right.$$

$$\left. + \nabla_w \mu_{k,w}(s_k) \nabla_{\xi_k} p(s_{k+1}|s_k, \xi_k) \Big|_{\xi_k = \mu_{k,w}(s_k)} V_{k+1}^{\pi_w, \psi_w}(s_{k+1}) \right] ds_{k+1}$$

$$= \nabla_w \mu_{k,w}(s_k) \nabla_{\xi_k} \left[ \int_{y_k} p(y_k|s_k, \xi_k) r_k^C(s_k, \xi_k, y_k) \, dy_k \right.$$

$$\left. + \int_{s_{k+1}} p(s_{k+1}|s_k, \xi_k) V_{k+1}^{\pi_w, \psi_w}(s_{k+1}) ds_{k+1} \right] \Big|_{\xi_k = \mu_{k,w}(s_k)}$$

$$+ \int_{s_{k+1}} p(s_{k+1}|s_k, \mu_{k,w}(s_k)) \nabla_w V_{k+1}^{\pi_w, \psi_w}(s_{k+1}) \, ds_{k+1}$$

$$= \nabla_w \mu_{k,w}(s_k) \nabla_{\xi_k} Q_k^{\pi_w, \psi_w}(s_k, \xi_k) \Big|_{\xi_k = \mu_{k,w}(s_k)}$$

$$+ \int_{s_{k+1}} p(s_k \to s_{k+1}|\pi_w) \nabla_w V_{k+1}^{\pi_w, \psi_w}(s_{k+1}) \, ds_{k+1}.$$

Combining the two terms, we have

$$\nabla_w V_k^{\pi_w, \psi_w}(s_k) = \mathbf{1}_{s_k \notin \mathcal{T}_{k,w}} \nabla_w \mu_{k,w}(s_k) \nabla_{\xi_k} Q_k^{\pi_w, \psi_w}(s_k, \xi_k) \Big|_{\xi_k = \mu_{k,w}(s_k)} \tag{43}$$

$$+ \mathbf{1}_{s_k \notin \mathcal{T}_{k,w}} \int_{s_{k+1}} p(s_k \to s_{k+1}|\pi_w, \psi_w) \nabla_w V_{k+1}^{\pi_w, \psi_w}(s_{k+1}) \, ds_{k+1}.$$

Applying the recursive (43) to itself repeatedly, we obtain

$$\nabla_w V_k^{\pi_w, \psi_w}(s_k)$$

$$= \mathbf{1}_{s_k \notin \mathcal{T}_{k,w}} \nabla_w \mu_{k,w}(s_k) \nabla_{\xi_k} Q_k^{\pi_w, \psi_w}(s_k, \xi_k) \Big|_{\xi_k = \mu_{k,w}(s_k)}$$

$$+ \mathbf{1}_{s_k \notin \mathcal{T}_{k,w}} \int_{s_{k+1}} p(s_k \to s_{k+1} | \pi_w, \psi_w)$$

$$\cdot \mathbf{1}_{s_{k+1} \notin \mathcal{T}_{k+1,w}} \nabla_w \mu_{k+1,w}(s_{k+1}) \nabla_{\xi_{k+1}} Q_{k+1}^{\pi_w, \psi_w}(s_{k+1}, \xi_{k+1}) \Big|_{\xi_{k+1} = \mu_{k+1,w}(s_{k+1})} ds_{k+1}$$

$$+ \mathbf{1}_{s_k \notin \mathcal{T}_{k,w}} \int_{s_{k+1}} p(s_k \to s_{k+1} | \pi_w, \psi_w) \mathbf{1}_{s_{k+1} \notin \mathcal{T}_{k+1,w}}$$

$$\cdot \int_{s_{k+2}} p(s_{k+1} \to s_{k+2} | \pi_w, \psi_w) \nabla_w V_{k+2}^{\pi_w, \psi_w}(s_{k+2}) \, ds_{k+2} \, ds_{k+1}$$

$$= \prod_{j=k}^{k} \mathbf{1}_{s_j \notin \mathcal{T}_{j,w}} \nabla_w \mu_{k,w}(s_k) \nabla_{\xi_k} Q_k^{\pi_w, \psi_w}(s_k, \xi_k) \Big|_{\xi_k = \mu_{k,w}(s_k)}$$

$$+ \int_{s_{k+1}} p(s_k \to s_{k+1} | \pi_w, \psi_w)$$

$$\cdot \prod_{j=k}^{k+1} \mathbf{1}_{s_j \notin \mathcal{T}_{j,w}} \nabla_w \mu_{k+1,w}(s_{k+1}) \nabla_{\xi_{k+1}} Q_{k+1}^{\pi_w, \psi_w}(s_{k+1}, \xi_{k+1}) \Big|_{\xi_{k+1} = \mu_{k+1,w}(s_{k+1})} ds_{k+1}$$

$$+ \int_{s_{k+2}} p(s_k \to s_{k+2} | \pi_w, \psi_w) \prod_{j=k}^{k+1} \mathbf{1}_{s_j \notin \mathcal{T}_{j,w}} \nabla_w V_{k+2}^{\pi_w, \psi_w}(s_{k+2}) \, ds_{k+2}$$

$$= \prod_{j=k}^{k} \mathbf{1}_{s_j \notin \mathcal{T}_{j,w}} \nabla_w \mu_{k,w}(s_k) \nabla_{\xi_k} Q_k^{\pi_w, \psi_w}(s_k, \xi_k) \Big|_{\xi_k = \mu_{k,w}(s_k)}$$

$$+ \int_{s_{k+1}} p(s_k \to s_{k+1} | \pi_w, \psi_w)$$

$$\cdot \prod_{j=k}^{k+1} \mathbf{1}_{s_j \notin \mathcal{T}_{j,w}} \nabla_w \mu_{k+1,w}(s_{k+1}) \nabla_{\xi_{k+1}} Q_{k+1}^{\pi_w, \psi_w}(s_{k+1}, \xi_{k+1}) \Big|_{\xi_{k+1} = \mu_{k+1,w}(s_{k+1})} ds_{k+1}$$

$$+ \int_{s_{k+2}} p(s_k \to s_{k+2} | \pi_w, \psi_w)$$

$$\cdot \prod_{j=k}^{k+2} \mathbf{1}_{s_j \notin \mathcal{T}_{j,w}} \nabla_w \mu_{k+2,w}(s_{k+2}) \nabla_{\xi_{k+2}} Q_{k+2}^{\pi_w, \psi_w}(s_{k+2}, \xi_{k+2}) \Big|_{\xi_{k+2} = \mu_{k+2,w}(s_{k+2})} ds_{k+2}$$

$$\vdots$$

$$+ \int_{s_N} p(s_k \to s_N | \pi_w, \psi_w) \prod_{j=k}^{N} \mathbf{1}_{s_j \notin \mathcal{T}_{j,w}} \nabla_w V_N^{\pi_w, \psi_w}(s_N) \, ds_N$$

$$= \sum_{l=k}^{N-1} \int_{s_l} p(s_k \to s_l | \pi_w, \psi_w) \prod_{j=k}^{l} \mathbf{1}_{s_j \notin \mathcal{T}_{j,w}} \nabla_w \mu_{l,w}(s_l) \nabla_{\xi_l} Q_l^{\pi_w, \psi_w}(s_l, \xi_l) \Big|_{\xi_l = \mu_{l,w}(s_l)} ds_l$$

$$= \sum_{l=k}^{N-1} \mathbb{E}_{s_l | \pi_w, \psi_w, s_k} \left[ \prod_{j=k}^{l} \mathbf{1}_{s_j \notin \mathcal{T}_{j,w}} \nabla_w \mu_{l,w}(s_l) \nabla_{\xi_l} Q_l^{\pi_w, \psi_w}(s_l, \xi_l) \Big|_{\xi_l = \mu_{l,w}(s_l)} \right] ds_l \tag{44}$$

At last, we obtain the policy gradient expression:

$$\nabla_w U(w) = \nabla_w V_0^{\pi_w, \psi_w}(s_0)$$

$$= \sum_{l=0}^{N-1} \mathbb{E}_{s_l | \pi_w, \psi_w, s_0} \left[ \prod_{j=0}^{l} (1 - \mathbf{1}_{s_j \in \mathcal{T}_{j,w}}) \nabla_w \mu_{l,w}(s_l) \nabla_{\xi_l} Q_l^{\pi_w, \psi_w}(s_l, \xi_l) \Big|_{\xi_l = \mu_{l,w}(s_l)} \right]$$

$$= \sum_{l=0}^{N-1} \mathbb{E}_{s_l | \pi_w, \psi_w, s_0} \left[ \mathbf{1}_{l < \tau_w} \nabla_w \mu_{l,w}(s_l) \nabla_{\xi_l} Q_l^{\pi_w, \psi_w}(s_l, \xi_l) \Big|_{\xi_l = \mu_{l,w}(s_l)} \right]. \tag{45}$$

$$\square$$

## A.4 IMPLEMENTATION DETAILS

This section addresses several practical challenges including neural network design for variable-length histories and efficient computation of information-theoretic rewards. Table 1 provides an overview of the key hyperparameters used in our implementation. We largely adopt parameter settings consistent with related work (Shen & Huan, 2023), and do not explore alternative architectures or tuning strategies here, as such choices are orthogonal to the main methodological contributions of the paper.

Table 1: Hyperparameter settings for numerical experiments.

|  | Linear Gaussian | Source Detection |
|---|---|---|
| Policy network architecture | $N + (N-1)(N_\xi + N_y) \to 80 \to 80 \to N_\xi$ | |
| Q-network architecture | $N + (N-1)(N_\xi + N_y) + N_\xi \to 80 \to 80 \to 1$ | |
| Learning rate (policy) | $1.5 \times 10^{-1}$ | |
| Learning rate (critic) | $1.0 \times 10^{-3}$ | |
| Critic batch size | 500 | |
| Exploration scale $\sigma_{\text{explore}}$ | 1.0 | 0.05 |
| Exploration scale decay | 0.99 | |
| MC size $M$ | 1000 | |

**Policy Network Architecture**   The policy network $\mu_w(k, s_k)$ takes as input both the current stage index and the experimental history. To handle the temporal nature of sequential experimentation, we design the input representation as follows: For the stage information, we employ one-hot encoding to capture the discrete temporal structure:

$$k \qquad \longrightarrow \qquad e_{k+1} = [0, \ldots, 0, \underbrace{1}_{(k+1)\text{th}}, 0, \ldots, 0]^T. \qquad (46)$$

for the second component, i.e., the state $s_k$ (including both $s_{k,b}$ and $s_{k,p}$), we represent it in a nonparametric manner

$$s_k \qquad \longrightarrow \qquad I_k = \{\xi_0, y_0, \ldots, \xi_{k-1}, y_{k-1}\}, \qquad (47)$$

which poses a challenge due to its variable length across different stages. We address this by using a fixed-size representation that accommodates the maximum possible history length. For experiments up to stage $(N-1)$, we allocate a total dimension of $(N-1)(N_\xi + N_y)$ and pad incomplete histories with zeros. The complete input vector for stage $k$ takes the form:

$$[\underbrace{e_{k+1}}_{N}, \overbrace{\xi_0}^{N_\xi}, \ldots, \xi_{k-1}, \underbrace{0, \ldots, 0}_{N_\xi(N-1-k)}, \overbrace{y_0}^{N_y}, \ldots, y_{k-1}, \underbrace{0, \ldots, 0}_{N_y(N-1-k)}]^T. \qquad (48)$$

This representation ensures consistent input dimensionality across all stages while preserving the sequential structure of the experimental process. The total input dimension scales linearly as $N + (N-1)(N_\xi + N_y)$. The output layer is an $N_\xi$-dimensional vector representing $\xi_k$, and the network architecture can be chosen by the user.

**Q-Network Architecture**   The Q-network follows a similar input design but includes an additional component for the candidate design $\xi_k$, resulting in input dimension $N + (N-1)(N_\xi + N_y) + N_\xi$. The network outputs a scalar value representing the expected future return.

**Information Gain Computation**   The terminal reward calculation requires computing KL divergences between posterior distributions. For problems without analytical posteriors, we approximate these quantities by discretizing the parameter space $\theta$ on a regular grid and estimating posterior densities pointwise. Figure 5 demonstrates the accuracy of this grid-based approximation in the linear-Gaussian case, where we compare the approximated posterior against the analytical solution. The close agreement validates that grid-based computation provides sufficient accuracy for reward evaluation in our experiments.

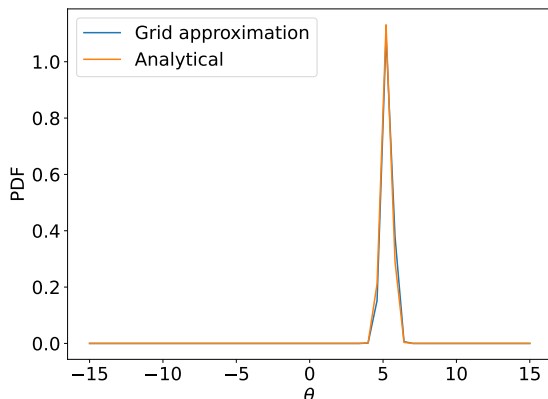

Figure 5: Comparison of grid-based posterior approximation against the analytical Gaussian posterior in the linear-Gaussian benchmark.

### A.5 ANALYTICAL SOLUTION TO THE LINEAR-GAUSSIAN PROBLEM

We derive the analytical solution to the linear-Gaussian problem described in section 5.1, under the terminal reward formulation. The conjugate Gaussian structure enables us to obtain closed-form expressions for the optimal stopping and design policies. Due to this conjugacy, both the prior and posterior distributions remain Gaussian throughout the sequential process, with log-density functions given by:

$$\ln p(\theta \mid I_0) = -\frac{1}{2}\ln(2\pi\sigma_0^2) - \frac{(m_0 - \theta)^2}{2\sigma_0^2} \tag{49}$$

$$\ln p(\theta \mid I_k) = -\frac{1}{2}\ln(2\pi\sigma_k^2) - \frac{(m_k - \theta)^2}{2\sigma_k^2}, \tag{50}$$

where $m_k$ and $\sigma_k^2$ denote the posterior mean and variance after performing $k$ experiments. These posterior parameters can be updated iteratively as:

$$(m_k, \sigma_k^2) = \left( \frac{\frac{y_{k-1}/\xi_{k-1}}{\sigma_\epsilon^2/\xi_{k-1}^2} + \frac{m_{k-1}}{\sigma_{k-1}^2}}{\frac{1}{\sigma_\epsilon^2/\xi_{k-1}^2} + \frac{1}{\sigma_{k-1}^2}}, \frac{1}{\frac{1}{\sigma_\epsilon^2/\xi_{k-1}^2} + \frac{1}{\sigma_{k-1}^2}} \right). \tag{51}$$

Alternatively, they can be computed directly from the complete experimental history:

$$\sigma_k^2 = \frac{\sigma_0^2 \sigma_\epsilon^2}{\sigma_\epsilon^2 + \sigma_0^2 \sum_{i=0}^{k-1} \xi_i^2}$$

$$m_k = \sigma_k^2 \left( \frac{m_0}{\sigma_0^2} + \frac{1}{\sigma_\epsilon^2} \sum_{i=0}^{k-1} y_i \xi_i \right). \tag{52}$$

For Gaussian distributions, the KL divergence has a closed form expression:

$$D_{\mathrm{KL}}(\mathcal{N}(m_a, \sigma_a^2) \,\|\, \mathcal{N}(m_b, \sigma_b^2)) = \frac{1}{2}\left[ \frac{\sigma_a^2}{\sigma_b^2} + \frac{(m_a - m_b)^2}{\sigma_b^2} + \ln \frac{\sigma_b^2}{\sigma_a^2} - 1 \right]. \tag{53}$$

This closed form allows us to derive explicit expressions for the expected information gain. Since $\sigma_{k+1}^2$ is updated independently of the observation $y_k$ for all $k$, the expected information gain from performing experiment $k$ is given by:

$$\mathbb{E}_{y_k|s_k,\xi_k}\left[ D_{\mathrm{KL}}\left( p_{\theta|I_{k+1}} \,\|\, p_{\theta|I_k} \right) \right]$$

$$= \mathbb{E}_{y_k|s_k,\xi_k}\left[ \frac{1}{2}\left\{ \frac{\sigma_{k+1}^2}{\sigma_k^2} + \frac{(m_{k+1} - m_k)^2}{\sigma_k^2} + \ln \frac{\sigma_k^2}{\sigma_{k+1}^2} - 1 \right\} \right]$$

$$= \frac{1}{2} \mathbb{E}_{y_k|s_k,\xi_k} \left[ \frac{(m_{k+1} - m_k)^2}{\sigma_k^2} \right] + \frac{1}{2} \left( \frac{\sigma_{k+1}^2}{\sigma_k^2} + \ln \frac{\sigma_k^2}{\sigma_{k+1}^2} - 1 \right)$$

$$= \frac{1}{2} \mathbb{E}_{y_k|s_k,\xi_k} \left[ \frac{\sigma_k^2 (y_k/\xi_k - m_k)^2}{(\sigma_k^2 + \sigma_\epsilon^2/\xi_k^2)^2} \right] + \frac{1}{2} \left( \frac{\sigma_{k+1}^2}{\sigma_k^2} + \ln \frac{\sigma_k^2}{\sigma_{k+1}^2} - 1 \right)$$

$$= \frac{1}{2} \frac{\sigma_k^2 \xi_k^2}{\sigma_k^2 \xi_k^2 + \sigma_\epsilon^2} + \frac{1}{2} \left( \frac{\sigma_{k+1}^2}{\sigma_k^2} + \ln \frac{\sigma_k^2}{\sigma_{k+1}^2} - 1 \right)$$

$$= \frac{1}{2} \left( 1 - \frac{\sigma_{k+1}^2}{\sigma_k^2} \right) + \frac{1}{2} \left( \frac{\sigma_{k+1}^2}{\sigma_k^2} + \ln \frac{\sigma_k^2}{\sigma_{k+1}^2} - 1 \right)$$

$$= \frac{1}{2} \ln \frac{\sigma_k^2}{\sigma_{k+1}^2} \tag{54}$$

in which computing the expectations utilizes the identities

$$\mathbb{E}_{y_k|m_k,\sigma_k^2,\xi_k} [y_k] = \int_{-\infty}^{+\infty} y_k p(y_k \mid m_k, \sigma_k^2, \xi_k) dy_k$$

$$= \int_{-\infty}^{+\infty} \int_{-\infty}^{+\infty} y_k p(y_k \mid \theta, m_k, \sigma_k^2, \xi_k) p(\theta \mid m_k, \sigma_k^2, \xi_k) dy_k d\theta$$

$$= \int_{-\infty}^{+\infty} \xi_k \theta p(\theta \mid m_k, \sigma_k^2, \xi_k) d\theta$$

$$= \xi_k m_k, \tag{55}$$

and

$$\mathbb{E}_{y_k|m_k,\sigma_k^2,\xi_k} \left[ y_k^2 \right] = \int_{-\infty}^{+\infty} y_k^2 p \left( y_k \mid m_k, \sigma_k^2, \xi_k \right) dy_k$$

$$= \int_{-\infty}^{+\infty} \int_{-\infty}^{+\infty} y_k^2 p \left( y_k \mid \theta, m_k, \sigma_k^2, \xi_k \right) p \left( \theta \mid m_k, \sigma_k^2, \xi_k \right) dy_k d\theta$$

$$= \int_{-\infty}^{+\infty} \left( \sigma_\epsilon^2 + \xi_k^2 \theta^2 \right) p \left( \theta \mid m_k, \sigma_k^2, \xi_k \right) d\theta$$

$$= \sigma_\epsilon^2 + \xi_k^2 \left( \sigma_k^2 + m_k^2 \right). \tag{56}$$

Using (54) we now present the first two steps in deriving the optimal sequential designs for the Linear-Gaussian case. For the last design,

$$\xi_{N-1}^* = \arg\max_{\xi_{N-1}} Q_{N-1}^{\pi,\psi}(s_{N-1}, \xi_{N-1})$$

$$= \arg\max_{\xi_{N-1}} \mathbb{E}_{y_{N-1}|\xi_{N-1}} \left[ V_N^{\pi,\psi}(s_N) \right]$$

$$= \arg\max_{\xi_{N-1}} \left\{ \mathbb{E}_{y_{N-1}|\xi_{N-1}} \left[ D_{\mathrm{KL}}(p_{\theta|I_N} \| p_{\theta|I_0}) \right] + \sum_{i=0}^{N-1} c_i(\xi_i) \right\}$$

$$= \arg\max_{\xi_{N-1}} \left\{ D_{\mathrm{KL}}(p_{\theta|I_{N-1}} \| p_{\theta|I_0}) + \sum_{i=0}^{N-2} c_i(\xi_i) \right.$$

$$\left. + \mathbb{E}_{y_{N-1}|\xi_{N-1}} \left[ D_{\mathrm{KL}}(p_{\theta|I_N} \| p_{\theta|I_{N-1}}) \right] + c_{N-1}(\xi_{N-1}) \right\}$$

$$= \arg\max_{\xi_{N-1}} \left\{ \mathbb{E}_{y_{N-1}|\xi_{N-1}} \left[ D_{\mathrm{KL}}(p_{\theta|I_N} \| p_{\theta|I_{N-1}}) \right] + c_{N-1}(\xi_{N-1}) \right\}$$

$$= \arg\max_{\xi_{N-1}} \left\{ \frac{1}{2} \ln \frac{\sigma_{N-1}^2}{\sigma_N^2} + c_{N-1}(\xi_{N-1}) \right\}$$

$$= \arg\max_{\xi_{N-1}} \left\{ \frac{1}{2} \ln \left[ \left( \frac{1}{\sigma_{N-1}^2} + \frac{\xi_{N-1}^2}{\sigma_\epsilon^2} \right) \sigma_{N-1}^2 \right] + c_{N-1}(\xi_{N-1}) \right\} \tag{57}$$

which achieves optimality at the maximum value $\xi_{N-1}^* = 3$ due to the monotone increasing property and design-independent costs. The corresponding maximum Q value is

$$Q_{N-1}^{\pi^*,\psi^*}(s_{N-1}, \xi_{N-1}^*) = D_{\mathrm{KL}}(p_{\theta|I_{N-1}} \,\|\, p_{\theta|I_0}) + \frac{1}{2}\ln\frac{\sigma_{N-1}^2}{\sigma_N^{*2}} + c_{N-1}(\xi_{N-1}^*). \tag{58}$$

The stopping set for this stage is then

$$\mathcal{T}_{N-1}^{\pi^*,\psi^*} = \left\{ s_{N-1} \mid r_T^S(s_{N-1}) \geq Q_{N-1}^{\pi^*,\psi^*}(s_{N-1}, \xi_{N-1}^*) \right\}$$

$$= \left\{ s_{N-1} \mid 0 \geq \frac{1}{2}\ln\frac{\sigma_{N-1}^2}{\sigma_N^{*2}} + c_{N-1}(\xi_{N-1}^*) \right\}, \tag{59}$$

which is independent of observations. Thus for stage $N-2$,

$$Q_{N-2}^{\pi,\psi}(s_{N-2}, \xi_{N-2}) = \mathbb{E}_{y_{N-2}|\xi_{N-2}} \max\left\{ r_T^S(s_{N-1}), Q_{N-1}^{\pi^*,\psi^*}(s_{N-1}, \xi_{N-1}^*) \right\}$$

$$= \max\left\{ \mathbb{E}_{y_{N-2}|\xi_{N-2}}\left[ r_T^S(s_{N-1}) \right], \mathbb{E}_{y_{N-2}|\xi_{N-2}}\left[ Q_{N-1}^{\pi^*,\psi^*}(s_{N-1}, \xi_{N-1}^*) \right] \right\}. \tag{60}$$

For the first case,

$$\xi_{N-2}^* = \arg\max Q_{N-2}^{\pi,\psi}(s_{N-2}, \xi_{N-2})$$

$$= \arg\max_{\xi_{N-2}} \left\{ \mathbb{E}_{y_{N-2}|\xi_{N-2}}\left[ D_{\mathrm{KL}}(p_{\theta|I_{N-1}} \,\|\, p_{\theta|I_0}) \right] + \sum_{i=0}^{N-2} c_i(\xi_i) \right\}$$

$$= \arg\max_{\xi_{N-2}} \left\{ \mathbb{E}_{y_{N-2}|\xi_{N-2}}\left[ D_{\mathrm{KL}}(p_{\theta|I_{N-1}} \,\|\, p_{\theta|I_{N-2}}) \right] + c_{N-2}(\xi_{N-2}) \right\}$$

$$= \arg\max_{\xi_{N-2}} \left\{ \frac{1}{2}\ln\left[ \left( \frac{1}{\sigma_{N-2}^2} + \frac{\xi_{N-2}^2}{\sigma_\epsilon^2} \right)\sigma_{N-2}^2 \right] + c_{N-2}(\xi_{N-2}) \right\}; \tag{61}$$

for the second case,

$$\xi_{N-2}^* = \arg\max Q_{N-2}^{\pi,\psi}(s_{N-2}, \xi_{N-2})$$

$$= \arg\max_{\xi_{N-2}} \left\{ \mathbb{E}_{y_{N-2}|\xi_{N-2}}\left[ D_{\mathrm{KL}}(p_{\theta|I_{N-1}} \,\|\, p_{\theta|I_0}) \right] + \frac{1}{2}\ln\left( \frac{\sigma_{N-1}^2}{\sigma_N^{*2}} \right) + c_{N-1}(\xi_{N-1}^*) \right\}$$

$$= \arg\max_{\xi_{N-2}} \left\{ \frac{1}{2}\ln\frac{\sigma_{N-2}^2}{\sigma_{N-1}^2} + \frac{1}{2}\ln\left( \frac{\sigma_{N-1}^2}{\sigma_N^{*2}} \right) + c_{N-2}(\xi_{N-2}) + c_{N-1}(\xi_{N-1}^*) \right\}$$

$$= \arg\max_{\xi_{N-2}} \left\{ \frac{1}{2}\ln\left[ \left( \frac{1}{\sigma_{N-2}^2} + \frac{\xi_{N-2}^2}{\sigma_\epsilon^2} + \frac{\xi_{N-1}^{*2}}{\sigma_\epsilon^2} \right)\sigma_{N-2}^2 \right] + c_{N-2}(\xi_{N-2}) \right\}. \tag{62}$$

Both cases are equivalent in achieving the optimality at the maximum value $\xi_{N-2}^* = 3$, due to the monotone increasing property and design-independent costs. Iteratively we have

$$\xi_k^* = \arg\max_{\xi_k} \frac{1}{2}\ln\left[ \left( \frac{1}{\sigma_k^2} + \frac{\xi_k^2}{\sigma_\epsilon^2} \right)\sigma_k^2 \right] = 3, \qquad \text{for } k = 0, \ldots, N-1. \tag{63}$$

The stopping sets for each stage are

$$\mathcal{T}_k^{\pi^*,\psi^*} = \left\{ s_k \mid 0 \geq \frac{1}{2}\ln\frac{\sigma_k^{*2}}{\sigma_{k+1}^{*2}} + c_k(\xi_k^*) \right\}$$

$$= \left\{ s_k \mid 0 \geq \frac{1}{2}\ln\left( 1 + \frac{\sigma_k^{*2}\sigma_0^2}{\sigma_\epsilon^2 + \sigma_0^2 \sum_{i=0}^{k-1} \xi_i^{*2}} \right) + c_k(\xi_k^*) \right\} \tag{64}$$

Finally the optimal utility for performing $N$ experiments is

$$Q_0^*(s_0) = \frac{1}{2}\ln\left[ \left( \frac{1}{\sigma_0^2} + \frac{\xi_0^2}{\sigma_\epsilon^2} + \frac{\xi_1^{*2}}{\sigma_\epsilon^2} + \ldots + \frac{\xi_{N-1}^{*2}}{\sigma_\epsilon^2} \right)\sigma_0^2 \right] + \sum_{i=0}^{N-1} c_i(\xi_i^*)$$

$$= \frac{1}{2}\ln\left( \sigma_\epsilon^2 + \sigma_0^2 \sum_{i=0}^{N-1} \xi_i^{*2} \right) - \frac{1}{2}\ln\sigma_\epsilon^2 + \sum_{i=0}^{N-1} c_i(\xi_i^*). \tag{65}$$

We list the optimal utilities for varying number of experiments in Table 2. The optimal stopping stages are highlighted in red.

Table 2: Optimal Utility for varying experimental horizons.

| $N$ | 1 | 2 | 3 | 4 |
|---|---|---|---|---|
| $U(\Xi_N)(c_k = 0)$ | 2.203 | 2.547 | 2.749 | 2.892 |
| $U(\Xi_N)(c_k = -0.5)$ | 1.703 | 1.547 | 1.249 | 0.892 |
| $U(\Xi_N)(c_k = -0.25)$ | 1.953 | 2.047 | 1.999 | 1.892 |

### A.6 NUMERICAL EXPERIMENTS DETAILS

#### A.6.1 STOPPING PROBABILITY SCHEDULE

To illustrate the role of the curriculum schedule, we compared several common choices for the stopping probability $p_{\text{stop}}(\ell)$ in Figure 6, including linear, exponential, and sigmoid increases. As shown in Figure 7, these schedules behave similarly on relatively simple problems such as the linear–Gaussian benchmark: all converge to the optimal policy, differing mainly in convergence patterns and speed. However, for more complex or long-horizon tasks such as the contaminant source detection problem, the choice of schedule becomes critical. As shown in Figure 8, a fast exponential schedule leads to insufficient exploration during early training, resulting in training instability and convergence to a suboptimal policy. In contrast, sigmoid and linear schedules with slower progression rates ensure adequate full-horizon exploration in early stages, leading to stable training and superior final performance. These results suggest that practitioners should tune the curriculum schedule based on problem complexity: simple problems tolerate aggressive schedules, while complex tasks benefit from gradual transitions that maintain exploration before allowing the optimal stopping rule to dominate training dynamics.

In our implementation, we adopt an adaptive sigmoid-based schedule to control the gradual increase of $p_{\text{stop}}(\ell)$. Specifically, we apply a shifted and scaled sigmoid function of the form:

$$p_{\text{stop}}(\ell) = \frac{1}{1 + \exp(-a(\ell - \ell_0))}, \tag{66}$$

where $a$ controls the transition steepness and $\ell_0$ determines the midpoint. We set parameters such that $p_{\text{stop}}(\ell)$ starts near zero, increases smoothly through mid-training, and saturates above 0.999 for the final 30 iterations to guarantee convergence to the optimal stopping policy. This sigmoid-based schedule provides a smooth transition that balances exploration during early training with stability near convergence, avoiding the abrupt changes that can destabilize actor-critic learning.

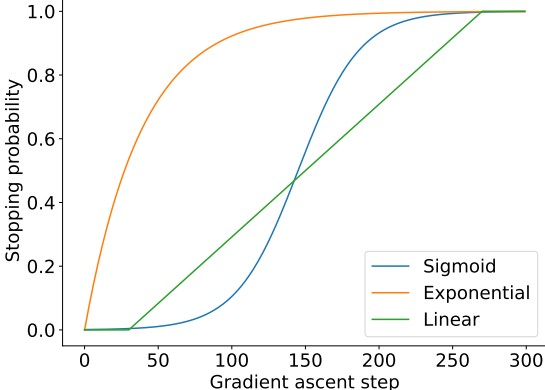

Figure 6: Examples of stopping probability schedules for curriculum PG over 300 training iterations.

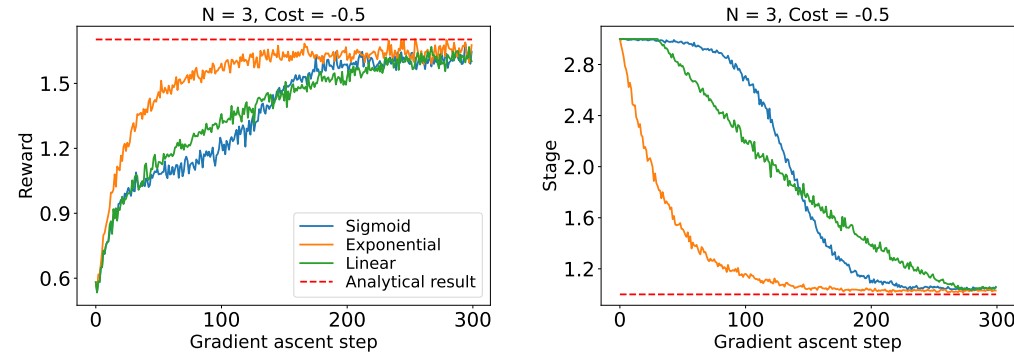

Figure 7: Comparison of different curriculum schedules on the Linear-Gaussian benchmark. Left: average reward; right: average stopping stage.

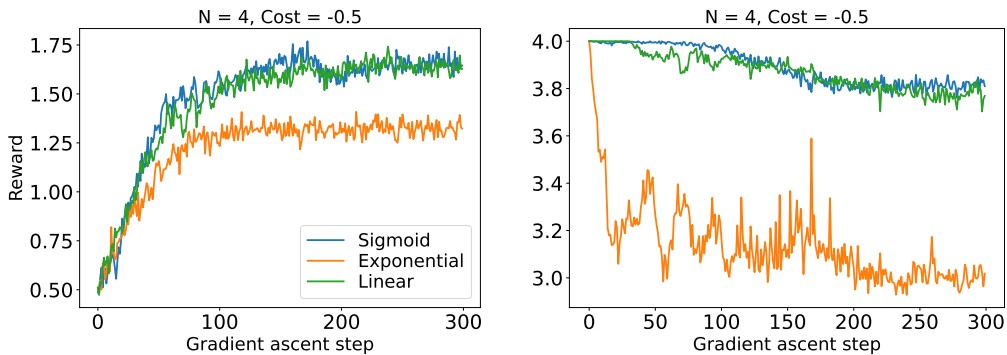

Figure 8: Comparison of different curriculum schedules on the convection-diffusion source detection problem. Left: average reward; right: average stopping stage.

### A.6.2 CONTAMINANT SOURCE DETECTION PROBLEM SETUP

The contaminant concentration $G$ at spatial location $z = [z_x, z_y]$ and time $t$ follows convection-diffusion PDE:

$$\frac{\partial G(z,t;\theta)}{\partial t} = \nabla^2 G - u(t) \cdot \nabla G + S(z,t;\theta),$$
$$z \in [z_L, z_R]^2, \quad t > 0 \tag{67}$$

where $\theta = [\theta_x, \theta_y, \theta_h, \theta_s] \in \mathbb{R}^4$ parameterizes the Gaussian source function

$$S(z,t;\theta) = \frac{\theta_s}{2\pi\theta_h^2} \exp\left(-\frac{(\theta_x - z_x)^2 + (\theta_y - z_y)^2}{2\theta_h^2}\right) \tag{68}$$

with location $(\theta_x, \theta_y)$, width $\theta_h$, and strength $\theta_s$. $u = [u_x, u_y] \in \mathbb{R}^2$ is a time-dependent convection velocity.

A mobile sensor measures noisy concentrations

$$y_k = G(z = s_{k+1,p}, t_k; \theta) + \epsilon_k \tag{69}$$

where $\epsilon_k \sim \mathcal{N}(0, \sigma_\epsilon^2)$. The design variable $\xi_k$ represents sensor displacement:

$$s_{k+1,p} = s_{k,p} + \xi_k. \tag{70}$$

We consider $N$ measurement opportunities at times $t_k = k \cdot \Delta t$ for $k = 0, \ldots, N-1$, with $\Delta t = 5.0 \times 10^{-4}$. We focus on source localization with $\theta_h = 0.05$, $\theta_s = 2$, $u_x = u_y = 10t/0.2$ and prior $\theta_x, \theta_y \sim \mathcal{U}(0,1)$.

Solving the forward model (67) using finite volume is still computationally expensive. We use DNNs to construct surrogate models of $G(z, t_k; \theta)$ for $k = 0, \ldots, N - 1$, to accelerate the computation. Each DNN uses a 4-dimensional input layer taking $z$ and $\theta$, 5 hidden layers with 40, 80, 40, 20, and 10 nodes, and a scalar output $G$. A dataset is generated by solving for $G$ on 2000 samples of $\theta$ drawn from its prior distribution. These concentration values are then first restricted to only the domain that is reachable by the vehicle (due to the design constraint), then shuffled across $\theta$ and split 80% for training and 20% for testing.

### A.6.3 Additional results

We present distributional analysis and representative episode comparisons to provide deeper insights into the learned policies.

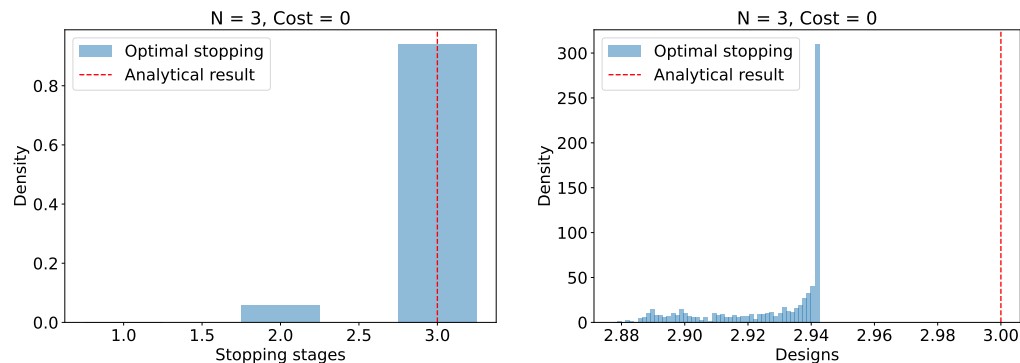

Figure 9: Distributional analysis of stopping stages (left) and experimental designs (right) for the linear-Gaussian benchmark with zero experimental cost ($c_k = 0$).

The distributional analysis in Figure 9 for the linear-Gaussian benchmark shows that the trained policy predominantly terminates at stage 3 with designs concentrated near the upper bound ($\xi = 3$), consistent with analytical solutions. The stopping stage histogram (left) confirms that the learned policy correctly identifies stage 3 as optimal when experimental costs are absent. The design histogram (right) reveals that optimal designs cluster near the constraint boundary, maximizing the signal-to-noise ratio by selecting the largest feasible design values. This concentration pattern validates that the policy gradient approach successfully learns the theoretically optimal behavior in this benchmark case.

Figure 10 presents representative episodes from both methods in the contaminant source detection problem ($c_k = -0.8$). Vanilla policy terminates after two experiments with a total reward of 1.33, while curriculum-trained policy continues for three experiments, achieving a higher total reward of 1.99. The curriculum approach achieves superior expected performance and more stable convergence, which is precisely what the optimization objective seeks to maximize.

Finally, we explore a more complex cost structure in the contaminant source detection problem with design-dependent experimental costs $c_k = -\|\xi_k\|^2$, where sensor movements to distant locations incur higher penalties. Figure 11 shows training convergence under this quadratic cost structure. Curriculum PG again outperforms the vanilla method, achieving significantly higher average rewards and converging to an optimal stopping stage around 4, while the vanilla PG prematurely stops around stage 3 due to underestimated continuation values. Figure 12 compares the spatial distribution of learned sensor movements under constant versus design-dependent costs across experimental stages. At stage $k = 0$, both policies select identical initial designs as they share the same initial state. However, as experiments progress, the two cost structures lead to markedly different exploration strategies. Under constant costs (upper panels), the policy selects diverse movements in later stages ($k = 2, 3$), freely relocating sensors across the spatial domain to maximize information gain without penalty. In contrast, under design-dependent costs (lower panels), the policy exhibits a distinctive pattern of movement selections that forms a curved trajectory from the origin, reflecting cautious exploration that balances information acquisition against the quadratic movement penalty $\|\xi_k\|^2$. This adaptive behavior demonstrates that our framework successfully incorporates spatially-

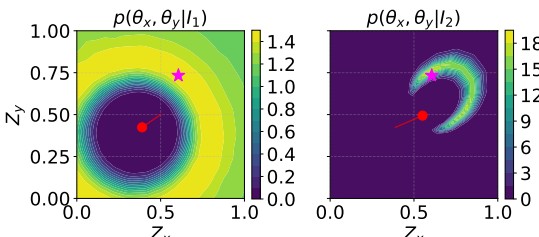

(a) Vanilla, $\theta = (0.61, 0.73)$, **total reward = 1.33** (information gain = 2.93, experimental cost = -1.6)

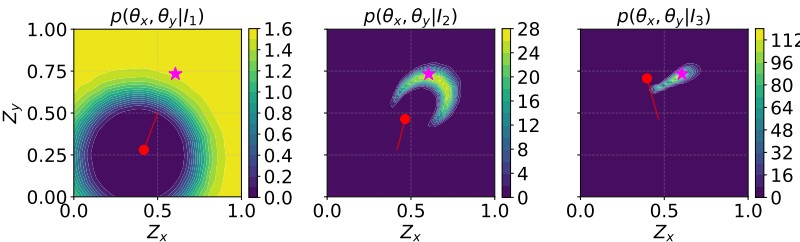

(b) Curriculum, $\theta = (0.61, 0.73)$, **total reward = 1.99** (information gain = 4.39, experimental cost = -2.4)

Figure 10: Episode instances for vanilla (upper) and curriculum (lower) policies in the $c_k = -0.8$ scenario. Purple star: true source location; red dots: sensor positions; red lines: sensor movements; contours: posterior PDF.

varying costs into the joint design and stopping optimization, learning policies that explicitly trade off exploration against experimentation costs.

These results demonstrate that the curriculum approach maintains its effectiveness even when cost structures become complex and spatially dependent, suggesting broader applicability to realistic experimental design problems where costs vary with design choices.

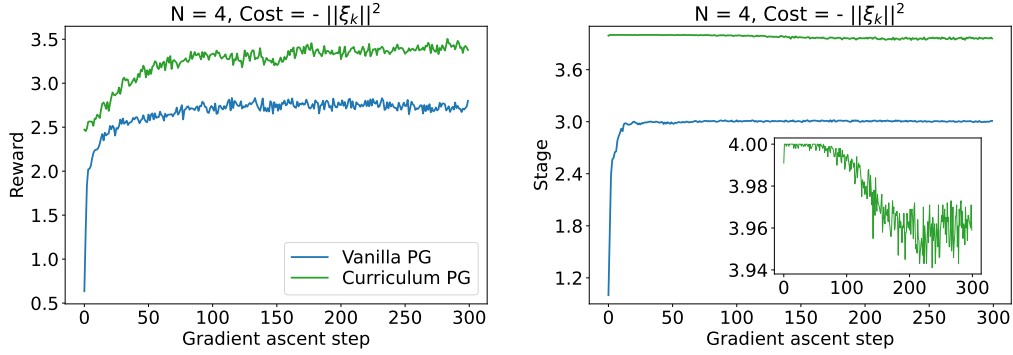

Figure 11: Training convergence of the convection-diffusion source detection problem with design-dependent costs $c_k = -\|\xi_k\|^2$. Left: average reward; right: average stopping stage.

## A.7 LLM ASSISTANCE DISCLOSURE

The authors acknowledge the use of Claude (Anthropic) and ChatGPT (OpenAI) for assistance in writing enhancement. All technical content, theoretical contributions, experimental results, and scientific insights remain entirely the work of the human authors. The AI assistance was limited to editorial and organizational improvements of existing author-generated content.

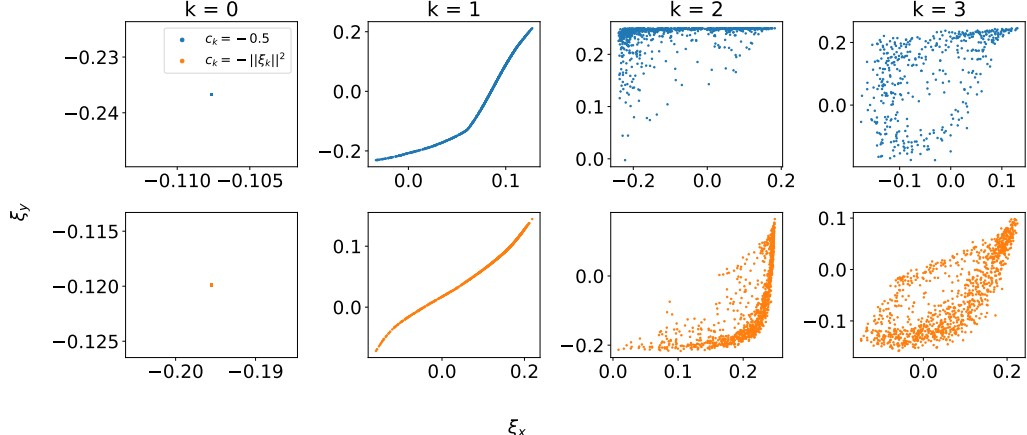

Figure 12: Comparison of learned sensor movement policies under constant versus design-dependent costs in the convection-diffusion source detection problem. Upper panels: constant cost $c_k = -0.5$. Lower panels: design-dependent cost $c_k = -||\xi_k||^2$.

