# OpenReview forum: "Optimal Stopping for Sequential Bayesian Experimental Design"
_ICLR.cc/2026/Conference — Submitted to ICLR 2026_

### Official Review · Reviewer_VivY · 2025-10-30

**Soundness:** 2
**Presentation:** 1
**Contribution:** 2
**Rating:** 4
**Confidence:** 4

**Summary:**

The paper presents a new approach to an understudied problem in Bayesian optimal experimental design (BOED) in the explicit likelihood setting that utilizes reinforcement learning (RL) policy-based BOED as opposed to variational inference (VI), non-RL policy-based, or maximum likelihood estimation of experimental designs and a critic (posterior) of interest. The new approach to early-stopping takes a simple rule of adding a constant loss to the "reward" in addition to the standard Expected Information Gain (EIG) in BOED. Results in two toy models of BOED are promising but lack evaluation in settings where the design isn't a constant cost.

**Strengths:**

The paper takes a good first approach to the optimal stopping problem in BOED through the lens of a MDP. I enjoyed reading about the connection between optimal stopping in the RL Bellman sense and optimal stopping in the setting of a constant cost. Allusion to some works in stochastic processes is mentioned but it seems RL is the most influential method, whereas I think more discussion of relation to stochastic processes would be beneficial. The authors also clearly state the intuition about how their framework works. They evaluate some of the most-critical hyperparameters of their method, demonstrating convergence to a known optimal stopping point for an analytic linear example.

**Weaknesses:**

The paper feels rushed. While the authors are addressing an interesting question in a narrow aspect of BOED, they drift into a pure RL formulation that sweeps more holistic evaluation of their method under a rug. BOED is based on the Bayesian principle of updating the current belief state of the posterior to prior. Their reward shares a similar achilles heel as other BOED methods in lacking evaluation of either the calibration of the resulting posterior or some form of validation metric outside of the EIG, which can be gamed. Omitting this type of evaluation makes me less confident in the results of the paper.

With respect to the method, I find it odd and arbitrary that the cost evaluated in the main paper is just a constant rather than a design-dependent value. For example, many real-world design depend on the experiment, such as the dollar amount of investing at a certain time. A constant seems to only penalize the time at which the experiment was performed. The paper partially addresses this in Figure 8 in the appendix but further analysis is omitted.

Additionally, the paper evaluates on only the source-finding problem in BOED and a trivial linear gaussian benchmark. Related to my previous critique, applying optimal stopping to a non-trivial problem that is design-dependent would demonstrate the utility of the paper's contribution.

Some of the exposition could be more clear, such as defining what the incremental utility $U_I$ are in Theorem 2 and tradeoffs between choosing it and $U_T$. Also, it'd help reading clarity to include those standalone symbols instead of mentioning them by name earlier then by symbol in the theorem.

I also have the following stand-alone critiques.
- Need to cite Kleinegesse et al. 2021 paper "Gradient-based Bayesian Experimental Design for Implicit Models using Mutual Information Lower Bounds" on value of future data points
- addictive -> additive at line 125, as well as other grammatical issues and typos throughout the paper
- It's not clear how the MDP formulation is unified in line 200. This seems like a broad and out-of-context statement.
- Simulation efficiency isn't discussed. According to Algorithm 1, we have L policy updates, each requiring M episodes, which uses N designs, so $L \times M \times N$ simulations, which, for real-world simulations, is quite expensive. I'd appreciate explicitly saying how many simulations were used in experiments.
- There doesn't seem to be a good analysis of the stopping probability choice, which would help practitioners implement this method.

**Questions:**

- How do the theorems change with design-dependent costs?
- Which form of mutual information are they defining the gradients w.r.t. to derive their gradient? It's not stated and makes the process of finding the gradient opaque, which concerns me.
- How does the exploration scale $\sigma_\text{explore}$ effect training? Why is it called $\epsilon_\text{explore}$ some places but not others? Assuming it's just the normal distribution parameterization making the designs follow a distribution but it'd be nice to make this more explicit.
- Is the design policy network parameterized by unique parameters at each step? i.e. $w_k$ that they use in their derivation in appendix A.3. Weird they use that as the gradient for deriving their optimal stopping point.
- How was the stopping probability schedule chosen? What are the implications of that choice?
- For the convection-diffusion model, Vanilla PG doesn't seem to underperform that dramatically. It does seem more conservative and maybe it has utility in other domains where design policies need to be more conservative? e.g. evaluating the EIG using CVAR.
- In line 105, what do the authors mean by RL can learn stopping rules but assume a fixed data acquisition process? Isn't this paper assuming a fixed data acquisition process (likelihood)?

---

> ### Author Response · Authors · 2025-11-23
> **Questions on Technical Details**
>
> We thank the reviewer for their detailed feedback. We address the questions and concerns below.
>
> **Q: How do the theorems change with design-dependent costs?**
>
> The theorems do not change; they hold for **arbitrary cost functions** $c_k(\xi_k)$, including design-dependent costs. Both the optimal stopping theorem (Theorem 1), the terminal-incremental equivalence (Theorem 2) and the policy gradient (Theorem 3) are derived based on the structure of the continuation value function and the Bellman optimality principle, without imposing any restrictions on the functional form of the cost.
>
> **Q: Which form of mutual information is used in the gradient derivations?**
>
> We used the expected KL divergence, as defined in the reward formulations (Eqs (4-7)). Importantly, the optimal stopping rule and policy-gradient derivation do not rely on properties specific to the KL form. The derivation applies to any scalar terminal utility, including alternative mutual-information metrics.
>
> **Q: Is the design policy network parameterized by unique parameters at each step?**
>
> In the policy-gradient derivation (Appendix A.3), we allow distinct parameters at each step, i.e., the gradient is taken with respect to $w = \\{w_0, …, w_{N-1}\\}$. In implementation, however, we use a single actor network shared across stages ($w_0 = … = w_{N-1}$), encoding the stage index as part of the input to distinguish stages. This simplifies training while preserving expressiveness. Additional implementation details are provided in Appendix A.4.
>
> **Q: How does the exploration scale affect training? Why is it mentioned in some places but not others?**
>
> The exploration scale controls the magnitude of a small perturbation added to the (deterministic) design policy when generating Monte Carlo trajectories for gradient estimation. This follows standard practice in actor–critic methods, where controlled noise during training prevents premature convergence to suboptimal designs before the critic has learned reliable continuation values.
> We apply this perturbation only during **MC rollout generation**; the policy itself remains deterministic, which is why exploration is discussed in the methods section but not elsewhere. We adopt a reasonable exploration schedule in (Shen & Huan, 2023), as systematically comparing exploration mechanisms is orthogonal to our main contributions.
>
> **Q: How was the stopping probability schedule chosen? What are the implications of that choice?**
>
> We thank the reviewer for this important practical question. The stopping probability schedule $p_{\text{stop}}(\ell)$ controls the curriculum learning process: it determines the probability of following the learned optimal stopping rule during training. The schedule should start at 0 to prevent premature stopping when continuation value estimates are unreliable, then gradually increase to 1 as the policy learns.
>
> We have added Appendix A.6.1 with detailed **guidelines on schedule selection** and a **comparison** of different schedule choices (linear, exponential, and sigmoid). Our analysis shows that:
> * **Simple problems** (e.g., linear-Gaussian benchmark): Different schedules behave similarly and all converge to the optimal policy, differing mainly in convergence speed.
> * **Complex problems** (e.g., contaminant source detection): Schedule choice becomes critical. Fast exponential schedules can lead to insufficient early exploration, causing training instability and convergence to suboptimal policies. Slower schedules (sigmoid or linear) ensure adequate fixed-horizon exploration during early training, resulting in stable learning and superior final performance.
>
> **Q: For the convection-diffusion model, vanilla PG doesn't underperform dramatically. Could this conservativeness be useful in risk-averse settings?**
>
> We thank the reviewer for this observation. The apparent "conservativeness" of vanilla policy-gradient actually arises from pessimistic continuation-value estimates during early training: a poorly initialized design policy undervalues future rewards, triggering premature stopping under the optimal stopping rule and preventing further improvement. This leads to suboptimal behavior rather than intentional risk aversion.
>
> If we were to evaluate EIG under a CVaR-based objective, similar underestimation of continuation values could cause the same premature-stopping behavior. Our curriculum approach explicitly mitigates this issue by stabilizing early training. Risk-aware variants of our method would require additional mechanisms to ensure continuation values accurately reflect risk preferences, an interesting direction for future work.

---

> ### Author Response · Authors · 2025-11-23
> **Weakness 2: Limited Problem Diversity**
>
> **Concern: The paper evaluates only two problems: a linear-Gaussian benchmark and source-finding.**
>
> We appreciate this feedback and would like to clarify the core contribution of our work. The primary contribution of this paper is the principled framework for optimal stopping in information-theoretic sequential BED, including:
> * The optimal stopping theorem (Theorem 1) that characterizes when to stop based on continuation values
> * The terminal–incremental equivalence (Theorem 2) extended to variable-horizon stopping settings
> * A practical policy-gradient algorithm that jointly learns design policies and stopping rules
>
> These theoretical and methodological contributions are broadly applicable to any sequential BED problem with stopping, regardless of the specific application domain.
>
> Our empirical evaluation was chosen strategically to validate these contributions:
> * Linear-Gaussian case: Provides rigorous validation against the analytical optimum, demonstrating that our framework correctly recovers the known optimal solution
> * Contaminant source detection: Demonstrates applicability to realistic, non-trivial problems with nonlinear PDE forward models, non-Gaussian priors, and complex posterior geometry
>
> The source-finding problem is representative of a broad class of inverse problems in environmental monitoring, geophysics, and epidemiology where stopping policies are practically relevant. While we agree that additional application domains, particularly with design-dependent costs, would further demonstrate empirical versatility, the generality of our framework does not depend on exhaustive empirical coverage. Our theoretical results and algorithmic contributions establish a foundation that practitioners can apply to their specific domains.
>
> We view expanding the empirical evaluation to additional problem classes as valuable future work that can build on the principled framework established in this paper.
>
> **Concern: The main experiments use constant costs rather than design-dependent costs.**
>
> We thank the reviewer for this point. Our theorems and method support arbitrary design-dependent cost functions $c_k(\xi_k)$, as clarified above. We chose constant costs in the main experiments for simplicity and clarity in demonstrating the stopping mechanism.
>
> We include design-dependent costs in Figure 11 (Appendix A.6.3), but agree that a more thorough analysis would strengthen the paper. We have expanded this section to provide **additional analysis** of how design-dependent costs affect the learned policies.
> While a comprehensive empirical study of how different cost structures affect policy behavior would be valuable, our primary contribution is the theoretical framework and algorithmic method that enables practitioners to incorporate arbitrary cost functions into their specific applications. The design-dependent cost example demonstrates the flexibility of our approach; exhaustive analysis across cost structures is beyond the scope of this work but represents a natural direction for future application-specific studies.

---

> ### Author Response · Authors · 2025-11-23
> **Other weakness and standalone critiques**
>
> **Weakness 1: The paper lacks evaluation of posterior calibration or validation metrics beyond EIG.**
>
> We appreciate this concern. We agree that evaluating learned designs through validation metrics beyond EIG is important for real-world deployment and represents a valuable direction for future research. Our focus in this work is on the reinforcement learning formulation of joint design and stopping within the information-theoretic BED framework, where EIG is the standard objective. Nevertheless, we acknowledge that additional validation metrics would strengthen the evaluation:
> * In the linear-Gaussian case, we validate against the analytical optimal design and stopping policies (Figure 9 in Appendix A.6.3), demonstrating correctness beyond EIG maximization
> * We have validated the accuracy of grid-approximation on the linear-Gaussian problem (Figure 5 in Appendix A.4), and such method is tractable for exact posterior evaluation for the problem dimensions we study ($N_\theta \le 2$)
>
> **Weakness 4: Clarity of Exposition**
>
> **Concern: Line 105 statement about "fixed data acquisition process" is confusing.**
>
> We appreciate the reviewer's question. Our intention was to distinguish prior RL-based stopping formulations that consider only a binary action space {continue, stop} without optimizing a design or control variable when continuing. We have revised the statement to:"Prior reinforcement learning formulations learn stopping policies over a binary action space {continue, stop}, without jointly optimizing the experimental design or control action taken when continuing."
>
> **Concern: Theorem 2 should define utilities more clearly.**
>
> We have added explicit equation references in Theorem 2 to clarify that the two utilities correspond to terminal-reward (Eqs (4) and (5)) and incremental-reward (Eqs (6) and (7)) formulations. We have also added a brief discussion in Section 4.2.3 on the **tradeoffs** between these formulations in implementation.
>
> **Related to standalone critiques**
>
> We thank the reviewer for the additional detailed feedback. We address each point below and have made corresponding revisions to the manuscript.
>
> **On literature**: We thank the reviewer for pointing out Kleinegesse et al. (2021) "Gradient-based Bayesian Experimental Design for Implicit Models using Mutual Information Lower Bounds." We have added this citation for future direction in optimal stopping for goal-oriented sequential experimental design.
>
> **On typos and grammatical issues**: We apologize for the typographical errors. We have carefully proofread the manuscript and corrected these errors throughout.
>
> **On the MDP formulation statement** (line 200): The eq (9) is the detailed objective function for the optimization problem. We have modified the sentence expression to improve clarity.
>
> **On sample efficiency**: We have incorporated a discussion on computation complexity in Section 4.4 and specified the settings of experiments in Appendix A.4.

---

### Official Review · Reviewer_7Jm2 · 2025-10-30

**Soundness:** 3
**Presentation:** 2
**Contribution:** 3
**Rating:** 6
**Confidence:** 3

**Summary:**

This paper formulates Bayesian experimental design as a joint design-and-stopping problem. The authors model sequential experiment selection as a belief-state MDP, where at each step the agent decides both which design (experiment) to perform and whether to stop collecting data. The paper proposes an actor-critic policy-gradient algorithm augmented with curriculum learning that jointly learns a design policy for experiment selection and a stopping policy for termination decisions.

**Strengths:**

The paper presents a clear and technically sound formulation of Bayesian experimental design as a joint design-and-stopping problem, explicitly modeling both experiment selection and termination decisions within a unified belief-state MDP.

While the general idea of comparing information gain to sampling cost has been explored in past work (e.g., knowledge gradient), this work is original in casting the problem as an explicit control process and learning both design and stopping policies jointly via a policy-gradient algorithm.

**Weaknesses:**

1. The paper can provide more discussions on related literature:
- The Knowledge Gradient (KG) policy [1,2] already incorporates a form of stopping rule for Bayesian experimental design, where sampling continues only while the expected value of information exceeds its cost. Although this stopping criterion is implicit rather than derived from a joint framework, it addresses a similar decision problem.
- The Pandora’s Box problem [3] is also a problem of joint selection and optimal stopping, where an agent sequentially explores costly options and decides when to stop — conceptually parallel to the structure of this paper’s MDP formulation.
- There is a recent work [4] also considers a joint selection and stopping framework, explicitly proposing a joint cost-aware policy for Bayesian optimization — a problem closely related to Bayesian experimental design.

2. The paper can provide brief explanations on terminologies such as curriculum learning, actor-critic, policy gradient for broader audiences and more reasons of these choices.

3. The paper can have more discussions on computational tractability and efficiency of their algorithms, particularly in high-dimensional continuous design space (there is one sentence in the conclusion but I'd like to see some explanations).

4. The paper can summarize all hyperparameters to be determined and sensitivity to choices.

References: \
[1] Frazier, Peter I., Warren B. Powell, and Savas Dayanik. "A knowledge-gradient policy for sequential information collection." SIAM Journal on Control and Optimization 47, no. 5 (2008): 2410-2439. \
[2] Ryzhov, Ilya O., Warren B. Powell, and Peter I. Frazier. "The knowledge gradient algorithm for a general class of online learning problems." Operations Research 60, no. 1 (2012): 180-195. \
[3] Weitzman, Martin. Optimal search for the best alternative. Vol. 78, no. 8. Department of Energy, 1978. \
[4] Xie, Qian, Linda Cai, Alexander Terenin, Peter I. Frazier, and Ziv Scully. "Cost-aware stopping for bayesian optimization." arXiv preprint arXiv:2507.12453 (2025).

**Questions:**

1. The paper’s formulation and stopping condition appear conceptually close to the Knowledge Gradient, which also incorporates a stopping rule in Bayesian experimental design. Could the authors clarify how their proposed policy differs from the KG approach? Could the authors elaborate on why policy-gradient policy can be more effective in this joint Bayesian experiment design and stopping problem?

2. The algorithm involves Monte Carlo simulation, is it time-consuming?

3. Could the authors clarify in what sense the learned policy is empirically optimal? For instance, is it approximately optimal with respect to the simulated belief dynamics, or are there any conditions under which convergence to the Bayesian-optimal solution can be expected? Can there be any potential model misspecification? How the approximations in Monte Carlo and policy-gradient optimization affect the optimality?

---

> ### Author Response · Authors · 2025-11-23
>
> ### **Weakness 1 & Question 1:  Related literature and comparison to Knowledge Gradient**
>
> We appreciate the reviewer’s thoughtful question regarding the relationship to Knowledge Gradient (KG).
>
> **On the difference from KG**: We note that Knowledge Gradient methods are not typically used in the information-theoretic BED domain; they originate from the Bayesian optimization and ranking-and-selection literature, where the objective is to identify the best alternative rather than to maximize information gain. Nevertheless, we can clarify the conceptual differences.
>
> Classical KG methods focus on **one-step lookahead** experiment selection under a fixed horizon and do not explicitly model the stopping decision. Some extensions introduce a cost-based criterion (e.g., stopping when the one-step KG value falls below the experiment cost), but this remains a **myopic** rule that compares only the immediate gain from the next experiment to its immediate cost.
>
> In contrast, our approach formulates the problem as a finite-horizon MDP and derives a stopping rule by comparing the terminal reward with the **full multi-step** continuation value (Theorem 1). This distinction becomes important in sequential settings where the value of continuing arises from the combined effect of multiple future experiments rather than a single step. Our policy-gradient method therefore supports a principled joint optimization of both design and stopping decisions, which KG-based heuristics do not aim to provide.
>
> **On why policy-gradient methods are effective**: Policy-gradient methods allow us to learn both (1) a flexible, continuous design policy $\pi(s)$ and (2) the continuation value $Q(s, \pi(s))$ needed for optimal stopping, in a unified framework. The continuation value captures long-horizon dependencies and enables non-myopic stopping decisions. In contrast, KG's one-step lookahead cannot account for the compounding value of future experiments, particularly in problems where information accumulates over multiple stages. Our experiments demonstrate that this non-myopic optimization substantially outperforms myopic stopping heuristics (Section 5).
>
> We have incorporated the suggested references into our related work section to provide broader context on stopping in adjacent domains, while clarifying that our contribution addresses stopping specifically within information-theoretic sequential BED, a setting where KG methods are not typically applied.
>
> ### **Weakness 3 & Question 2:  Discussion on computational tractability**
>
> We appreciate the reviewer’s question. We have incorporated more discussion in the revised manuscript.
>
> The policy gradient method requires Monte Carlo trajectory simulation during training to estimate gradients. The primary computational expense comes from evaluating information-theoretic rewards (KL divergences between posteriors) and forward model evaluations. For low-dimensional parameter spaces ($N_\theta \le 4$), we employ grid-based discretization to compute posteriors and KL divergences, which is both efficient and transparent. For higher-dimensional problems, practitioners can leverage well-established approximation methods such as variational inference, MCMC-based density estimation, or amortized posterior inference via diffusion models, depending on problem structure and available computational resources.
>
> Once trained, the policy networks enable computationally efficient decision-making. At deployment time, determining the next design and stopping decision requires only a single forward pass through the policy and Q-networks, plus one posterior update and forward model evaluation, eliminating the need for expensive online planning or repeated forward simulations. This efficiency makes the learned policy practical for real-time sequential experimentation, even in resource-constrained settings.
>
> ### **Weakness 2, 4:  Explanation on terminologies and hyperparameters settings**
>
> We thank the reviewer’s suggestion. In the revised manuscript we have incorporated the hyperparameter settings in Appendix A.4 and provided brief explanations on terminologies including curriculum learning, policy gradient and actor-critic.

---

> ### Author Response · Authors · 2025-11-23
>
> ### **Question 3: On optimality and approximation guarantees**
>
> We thank the reviewer for raising this important point regarding the sense in which our learned policy is optimal.
>
> **On the optimality of the learned policy**: The learned policy is trained to maximize expected return under the simulated belief dynamics and reward structure defined by our MDP formulation. In the linear–Gaussian example, where the Bayesian-optimal stopping boundary is available in closed form, we observe close agreement between the learned policy and the ground-truth optimal policy (Figure 9, Appendix A.6.3), providing direct validation that our method can recover the optimal solution when it is known.
>
> **On convergence guarantees**: For general nonlinear models, our method is an approximate actor–critic algorithm and therefore does not provide universal guarantees of convergence to the Bayesian-optimal policy. This limitation is shared by essentially all deep RL methods applied to continuous belief spaces. Convergence to the optimal solution is theoretically expected only under classical conditions (e.g., exact value function representations, tabular settings with sufficient exploration, or convex policy classes) that do not hold in our high-dimensional belief state setting.
>
> **On model misspecification**: Our method assumes the likelihood or simulator used during training correctly represents the true data-generating process, as is standard in Bayesian experimental design. If the model is misspecified, the learned policy optimizes for the assumed model rather than the true system and may deviate from the true optimal policy. Robustness to model misspecification is an important direction for future work but is beyond the scope of this paper.
>
> **On approximation errors**: Monte Carlo estimators and policy-gradient updates introduce variance into the learning process. However, these errors diminish with increased sample sizes and training iterations. In practice, we find that the learned policies are stable across multiple training runs and consistently outperform baseline heuristics, suggesting that the approximation quality is sufficient for practical performance.

---

> > ### Comment · Reviewer_7Jm2 · 2025-11-26
> >
> > Thanks for the reviewers' detailed responses. I appreciate you incorporating many of my concerns in the revised manuscript. However, I still have some concerns:
> >
> > 1. There are KG papers with objective of maximizing information gain rather than identifying the best alternative. One example is:
> >
> > [1] Frazier, Peter I., Warren B. Powell, and Savas Dayanik. "A knowledge-gradient policy for sequential information collection." SIAM Journal on Control and Optimization 47, no. 5 (2008): 2410-2439.
> >
> > KL seems to be just one special case of "value of information". Looks like KG can also be applied to the setting you considered.
> >
> > Also, in your linear–Gaussian example, KL divergence is strictly non-decreasing and you also mentioned that the optimal policy is to continue until the end of the horizon. However, this makes me think whether KL is an appropriate or informative objective for the stopping formulation you propose.
> >
> > 2. I noticed you provided timing results to other reviewers. However, it would be more convincing that your method is efficient if you can compare with some "baselines".

---

> ### Author Response · Authors · 2025-11-30
> **Response to Follow-up Comments**
>
> We thank the reviewer for the continued engagement and address the concerns below.
>
> **1a: The cited paper does NOT maximize information gain.**
>
> The reviewer claims "there are KG papers with objective of maximizing information gain." This is factually incorrect for the cited paper. Frazier et al. (2008) addresses **ranking and selection with a fixed budget**, identifying which of K finite alternatives has the highest mean value. The objective is to maximize expected decision value, not information gain. The phrase "information collection" in the title refers colloquially to data gathering for decision-making, not information theory.
>
> Our work addresses a fundamentally different problem: information-theoretic sequential experimental design with endogenous optimal stopping, where the objective is to maximize KL divergence (information gain) subject to experimental costs.
>
> The reviewer's claim that "KG can also be applied to the setting you considered" is not accurate. KG addresses ranking and selection with decision-theoretic objectives and fixed budgets. We address information gain maximization for continuous parameters with optimal stopping. These are different problem classes.
>
> **1b: KL divergence IS the appropriate objective for our stopping formulation.**
>
> The reviewer suggests that because KL is non-decreasing in the linear-Gaussian case, it is "not appropriate or informative" for stopping. This misunderstands the role of experimental costs and our validation strategy.
>
> **With zero experimental cost**: The expected KL is non-decreasing, so the optimal stopping policy is analytically known to continue until the maximum horizon N. Our method correctly learns this behavior in both the linear-Gaussian case and the source detection problem, validating algorithmic correctness.
>
> **With non-zero experimental costs**: The optimal stopping policy must be learned by balancing future information gain against costs. Even though KL may be non-decreasing, continuing becomes suboptimal when the expected future gain cannot cover the experimental costs.
>
> The monotonicity of KL with zero cost provides a known ground truth for validation. Non-zero costs make stopping non-trivial and practically essential. While we focus on KL divergence (the standard information-theoretic criterion in BED), our framework is agnostic to the specific form of information gain and can readily adapt to other value-of-information metrics.
>
> **2: Efficiency comparison with baselines**
>
> There are no existing baselines for optimal stopping in information-theoretic sequential BED. As we detailed in our response to reviewer PSxG, existing policy-based BED methods (Huan & Marzouk 2016, Foster et al. 2021) were designed for fixed-horizon problems and cannot handle endogenous stopping without fundamental architectural redesign, including redefining value functions, terminal conditions, and trajectory generation. Knowledge Gradient methods address ranking-and-selection, not information-theoretic BED.
>
> Our timing results demonstrate computational practicality, not superiority over non-existent baselines. Training is tractable on standard hardware, and once trained, deployment requires only single forward passes through the policy networks, enabling real-time decision-making without expensive online planning or repeated simulations.
>
> To nevertheless provide meaningful empirical validation, we pursue two complementary strategies:
> * **Comparison against analytical optimum** (linear–Gaussian case): We demonstrate that our method recovers the known optimal solution, establishing correctness and convergence.
> * **Comparison against naïve stopping heuristics**: We evaluate against several intuitive alternatives that practitioners might adopt, including threshold-based rules (stop when accumulated reward reaches a threshold) and fixed-stage stopping rules. While straightforward to implement, these heuristics do not account for future value and cannot adapt to the evolving trade-off between information gain and experimentation cost. Our method achieves consistently higher expected return across all tasks, demonstrating both the necessity of a principled stopping rule and establishing the first viable baseline for this problem class.

---

### Official Review · Reviewer_PpyM · 2025-11-01

**Soundness:** 2
**Presentation:** 2
**Contribution:** 2
**Rating:** 4
**Confidence:** 3

**Summary:**

The paper frames sequential Bayesian experimental design with an explicit stop action. It proves that the optimal rule is to stop when the terminal reward exceeds the expected continuation value, and shows that terminal and incremental reward formulations are equivalent. It then trains a policy-gradient actor–critic: the actor proposes designs, the critic estimates continuation value (Q), and stopping is decided by comparing Q to the terminal reward. To avoid early collapse, it introduces a curriculum schedule that gradually unlocks stopping during training. Experiments on a linear–Gaussian benchmark and a convection–diffusion (PDE) contaminant-source task show that the curriculum stabilizes training and reduces premature stopping compared to a vanilla PG baseline.

**Strengths:**

1. The paper makes stopping explicit and principled: decisions follow a simple rule—stop when the terminal reward meets or exceeds the continuation value—so it’s easy to implement and explain.

2. The theory reduces stopping to a concrete test (terminal reward vs. the critic’s Q), making the decision operational rather than heuristic.

3. Training is simple and effective: an actor–critic with a lightweight curriculum that prevents early collapse and stabilizes joint learning.

4. Empirically, the curriculum improves stability and cuts premature stopping on both the linear–Gaussian benchmark and the PDE contaminant-source task compared with a vanilla PG baseline.

**Weaknesses:**

1. The terminal–incremental equivalence (Theorem 2) is already established in DAD; this paper mainly adds an experimental-cost term. The authors should acknowledge this more clearly and tighten the novelty claim.

2. When N = 4, directly approximating the posterior (e.g., variational inference, Gaussian approximations, or diffusion-based posteriors) and then computing the reward may be cheaper and clearer than training an actor–critic. The paper should justify when learning actually outperforms these exact/analytic or grid-based methods.

3. DAD/iDAD/Step-DAD [1] handle non-Gaussian and implicit models, while this paper only shows Gaussian cases. If everything is Gaussian, closed-form posteriors/EIG are available—so the paper should exploit those baselines and/or include at least one non-Gaussian setting to justify the scope.

4. The stopping schedule is hand-designed, not learned; results may be schedule-dependent, and there’s little ablation on the shape, duration, or final gating level.



[1] Hedman, Marcel, et al. "Step-DAD: Semi-Amortized Policy-Based Bayesian Experimental Design." arXiv preprint arXiv:2507.14057 (2025).

**Questions:**

1. Have you applied your approach with alternative posterior approximations (e.g., variational inference, Gaussian, diffusion-based posteriors) and compared the results within the same framework? If not, what adaptations would be needed?

2. Did you try alternatives to stop-gating (e.g., a small per-step delay penalty, scheduled sampling, or entropy regularization on "continue")? If not, what was the intuition for choosing curriculum?

3. The current experiments assume Gaussian priors and Gaussian noise. Have you evaluated a Laplace approximation to the posterior as a baseline?

4. Please report wall-clock and sample-efficiency numbers (training time, number of episodes) to assess practical performance.

---

> ### Author Response · Authors · 2025-11-23
>
> ### **Weakness 1:  On novelty of Theorem 2: Terminal–incremental equivalence with stopping**
>
> We thank the reviewer for this comment. We agree that DAD (Foster et al., 2021) (and Shen & Huan (2023)) establish a terminal–incremental equivalence, but their result applies only to fixed-horizon experimental design, where the accumulated reward is summed across a fixed number of experiments.
>
> Our setting differs fundamentally: the summation length is **variable** and determined endogenously by the policy through a stopping rule. Extending the equivalence to this setting is not a direct corollary of prior work, because the return must be evaluated at a **random stopping time** $\tau \leq N$, and the terminal reward must be defined to match the cumulative incremental information gained up to that $\tau$. Establishing that the terminal and incremental formulations produce the **same optimal stopping rule** requires additional analysis that is absent in fixed-horizon formulations. Thus, our contribution is not simply adding an experimental-cost term, but generalizing the terminal–incremental equivalence to a policy-dependent stopping framework.
>
> We have revised the manuscript to highlight this distinction and emphasize the novelty of extending terminal–incremental equivalence to variable-horizon stopping.
>
>
>
> ### **Weakness 2 & Question 1: Posterior approximation methods and actor-critic necessity**
>
> We thank the reviewer for this suggestion, but we believe there may be a misunderstanding about the role of the actor-critic framework in our method.
>
> The actor-critic framework is not used for posterior approximation. Methods such as variational inference, Gaussian approximations, or diffusion-based posteriors could serve as alternatives to our grid-based computation of expected information gain, but these would only change how the per-step reward is evaluated. The core challenge in our setting is optimal sequential decision-making under an endogenous stopping rule, not posterior inference.
>
> Specifically, the actor-critic framework learns the **continuation value function**, which represents the expected future information gain conditional on the policy possibly stopping at any future stage. This continuation value is essential for determining the optimal stopping rule and cannot be obtained from posterior approximations alone. Even if the posterior were computed analytically or via advanced approximation methods, one would still need a mechanism, such as our critic, to estimate future value under variable-horizon trajectories and produce a consistent stopping policy.
>
> **On alternative posterior approximations**: Our framework is compatible with any posterior approximation method capable of providing (i) posterior predictive samples and (ii) posterior updates after observations. Methods such as variational inference, Gaussian approximations, or diffusion-based posteriors could be incorporated if they improve reward computation accuracy. However, we have not explored these alternatives in this work because:
> * The choice of posterior approximation affects only the accuracy of the EIG reward, not the learning dynamics of the design or stopping policies
> * Our focus is on the RL formulation of joint design and stopping, rather than on posterior modeling techniques
> * Grid-based computation is tractable exact EIG evaluation for the problem dimensions we study ($N_\theta \leq 2$)
>
> Exploring and benchmarking alternative posterior approximations would be a valuable direction for future work, particularly for higher-dimensional problems, but it is orthogonal to our core contribution of establishing the optimal stopping framework for information-theoretic BED.

---

> ### Author Response · Authors · 2025-11-23
>
> ### **Weakness 3 & Question 3: On non-Gaussian case studies**
>
> We would like to clarify a potential misunderstanding: the contaminant source detection problem presented in Section 5.2 is indeed a **non-Gaussian, nonlinear** setting. The non-Gaussian nature arises from two sources: 1) We use a uniform prior on the source parameters $\theta$, and 2) The forward model is a non-linear PDE (Eq. 67 in Appendix A.6.2), which induces a highly nonlinear parameter-observation relationship. Consequently, the posterior $p(\theta|y)$ is complex and intractable, and no closed-form expressions exist for either the posterior or the expected information gain.
>
> **On the linear-Gaussian case**: In this setting, the posterior is indeed available analytically, making a Laplace approximation unnecessary (where the posterior is **exactly Gaussian**). We use this case as a validation benchmark precisely because the analytical solution is available: it allows us to verify that our method correctly recovers the optimal design and stopping policies.
>
> **Regarding posterior approximation methods**: while posterior inference is necessary for computing the EIG reward, it is not the core focus of our work—our contribution is learning optimal design and stopping policies. In Figure 5 of Appendix A.4, we compare our grid-based posterior approximation against the analytical posterior to demonstrate that it provides sufficient accuracy for reward computation in this tractable setting.
>
> ### **Weakness 4 & Question 2: implication of curriculum schedule and alternatives**
>
> We appreciate the reviewer’s question regarding alternatives to stop-gating.
>
> The stopping probability schedule $p_{\text{stop}}(\ell)$ controls the curriculum learning process: it determines the probability of following the learned optimal stopping rule during training. The schedule should start at 0 to prevent premature stopping when continuation value estimates are unreliable, then gradually increase to 1 as the policy learns.
> We have added Appendix A.6.1 with detailed **guidelines on schedule selection** and a **comparison** of different schedule choices (linear, exponential, and sigmoid). Our analysis shows that:
> * **Simple problems** (e.g., linear-Gaussian benchmark): Different schedules behave similarly and all converge to the optimal policy, differing mainly in convergence speed.
> * **Complex problems** (e.g., contaminant source detection): Schedule choice becomes critical. Fast exponential schedules can lead to insufficient early exploration, causing training instability and convergence to suboptimal policies. Slower schedules (sigmoid or linear) ensure adequate fixed-horizon exploration during early training, resulting in stable learning and superior final performance.
>
> **Regarding alternative suggestions**: A per-step delay penalty would change the reward structure and could lead to biased policy. Entropy regularization is primarily applicable to stochastic policies; since our design policy is deterministic (with only exploration noise added), entropy regularization does not meaningfully mitigate premature stopping.
>
> ### **Question 4: report wall-clock and sample-efficiency numbers**
> We report wall-clock times for both experiments in the table. Training times correspond to **300** gradient ascent steps with **1000** Monte Carlo samples per step, executed on a MacBook Pro [M3, 36 GB RAM] using CPU. Deployment time is per decision (single forward pass through policy and Q-networks, plus one posterior update).}
>
> | Experiment                     | **Training Time (min)** | **Deployment Time (ms)** |
> |-------------------------------|--------------------------|---------------------------|
> | Linear-Gaussian (N=4)         | 9.3                      | 14.4                      |
> | Contaminant Detection (N=4)   | 156.5                    | 26.9                      |

---

### Official Review · Reviewer_PSxG · 2025-11-03

**Soundness:** 1
**Presentation:** 2
**Contribution:** 1
**Rating:** 2
**Confidence:** 4

**Summary:**

The authors suggest that choosing when to stop running new experiments is a neglected problem within Bayesian experimental design. They propose solving this using a policy-based approach in which the action space of the experimental-design policy includes “stop”. Their training method is an actor-critic algorithm in which stochasticity in the stopping decision is reduced over training steps (they refer to this as “curriculum learning”). They report training results for their method and some alternative methods.

**Strengths:**

Originality: popular policy-based approaches to BED (eg, Foster et al, 2021; Huan & Marzouk, 2016) tend to assume a sequence of T experiments; this work considers stopping after fewer than T experiments.

Quality: the writing is generally good, and I’m not aware of any major technical errors.

Clarity: aside from an overload on equations in Sections 3-4, the paper is reasonably easy to follow.

Significance: incorporating stopping into policy-based BED is a good idea for the future work to incorporate.

---

Foster et al (2021). Deep adaptive design: amortizing sequential Bayesian experimental design. ICML.

Huan & Marzouk (2016). Sequential Bayesian optimal experimental design via approximate dynamic programming. arXiv.

**Weaknesses:**

I’m currently unconvinced of how substantive and necessary the contribution is, and I worry that the work is presented in a way that obfuscates the nature of the contribution.

The authors tell us the BED literature has neglected stopping. Yet there is actually a good amount of coverage, including in work by Berger (1985), Garnett (2023) and Ryan et al (2016).

The authors also present three equation-heavy pages of methodological description, with the implication is that working out a solution requires this level of complexity. But it is unclear to me that the method doesn’t just amount to including “stop” in the action space of the experimental-design policy, with which we can then just use an existing algorithm, such as that of Huan & Marzouk (2016) without any modifications, or that of Foster et al (2021) with minor modifications to account for experiment costs.

On top of this, the work does not match the machine-learning community’s typical standards for evaluating new methods. The authors present training results, not test results. They also do not compare against notable methods from past work, such as those mentioned above.

---

Berger (1985). Statistical Decision Theory and Bayesian Analysis. Springer.

Garnett (2023). Bayesian Optimization. Cambridge University Press.

Ryan et al (2016). A review of modern computational algorithms for Bayesian optimal design. International Statistical Review.

**Questions:**

Can you explain why you can’t just augment existing policy-based BED, such as the method proposed by Huan & Marzouk (2016), to include “stop” in the action space of the experimental-design policy?

How well does your method perform in testing rather than training?

How well do the abovementioned baseline methods work?

---

> ### Author Response · Authors · 2025-11-23
>
> ### **Weakness 1:  Related literature in BED**
>
> We appreciate the reviewer’s comment and have carefully revisited the cited works. While these papers indeed study various forms of stopping, they address different problems from **information-theoretic** sequential Bayesian experimental design.
>
> * Berger (1985) studies Bayesian decision-theoretic stopping rules for determining when to stop sampling in order to minimize expected decision loss. This setting does not involve design selection or information-gain objectives, and the optimality principle derives from decision-loss tradeoffs rather expected information gain (EIG).
> * Garnett (2023) discusses early stopping criteria in Bayesian optimization, where the objective is to maximize an unknown function and stopping is governed by regret or improvement criteria. This literature focuses on global optimization rather than information-theoretic experimental design, and does not consider EIG-based utilities or sequential design of experiments.
> * Regarding Ryan et al. (2016), we were unable to identify a stopping rule or an endogenous stopping formulation in that work. The paper focuses on Bayesian utility functions and methods for their estimation, but the experimental horizon is fixed and stopping is not part of the design problem.
>
> In contrast, our work studies optimal stopping for information-theoretic sequential Bayesian experimental design, where the goal is to maximize expected information gain subject to an endogenous stopping time that must be learned jointly with the design policy. To the best of our knowledge, this specific combination — **EIG-based** sequential design with an optimal stopping rule derived and learned jointly — has not been directly addressed in prior **information-theoretic** BED literature.
>
> However, we have incorporated these works into our related work section to acknowledge the important developments of stopping in these other domains.
>
>
> ### **Weakness 2 & Question 1: the necessity of our framework and why existing methods cannot trivially incorporate stopping**
>
> We appreciate the reviewer’s intuition that stopping might be incorporated by simply adding a discrete "stop" action to existing policy-based BED algorithms. However, this is not a minor modification, and we clarify below why our framework is necessary.
> * First, the mixed discrete-continuous action space introduces fundamental computational challenges. Augmenting a continuous design action space with a discrete stop action introduces **nondifferentiability** into the policy class. Methods such as Foster et al. (2021) and Shen & Huan (2023) rely on fully differentiable, continuous design policies; mixing a discrete action with a continuous action invalidates their gradient estimators or requires substantial redesign of both the policy and the critic architectures. In contrast, our formulation avoids this issue entirely: the stopping decision is not parameterized as part of the policy class but is instead derived analytically with the optimal stopping theorem, leaving the design policy fully differentiable.
> * Second, early termination fundamentally changes the structure of the sequential decision problem. The agent no longer receives an information-gain reward at each stage, and the state does not necessarily transition to the next experimental posterior. Instead, the reward must include a state-dependent terminal value summarizing the accumulated information, and the transition must accommodate terminating trajectories. These changes require rebuilding the **Bellman recursion** from the ground up, rather than simply adding experiment costs to the reward.
>
> Consequently, whether for dynamic-programming-based methods with implicitly represented policies (Huan & Marzouk, 2016) or for policy-gradient methods (Shen & Huan, 2023; Foster et al., 2021), adapting them to the stopping setting requires major structural and computational changes. This includes redefining value functions, terminal conditions, trajectory generation procedures, and learning algorithms. Our framework provides a principled solution to these challenges by analytically characterizing the optimal stopping rule and decoupling it from the design policy, which both simplifies the problem structure and maintains computational tractability.

---

> ### Author Response · Authors · 2025-11-23
>
> ### **Weakness 3 & Questions 2, 3: Testing results and baseline comparisons**
>
> We thank the reviewer for raising these important evaluation question.
>
> **On test performance**: Test performance is evaluated using Monte Carlo roll outs without design perturbations and without the curriculum schedule. These conditions coincide with the late stages of training, where the exploration scale has decayed to zero and the curriculum probability has approached one. As a result, the behavior of the learned policy near convergence is already representative of its test performance. Reporting separate "test" metrics would duplicate what is already shown in the converged training behavior and would not provide additional insight into the role of curriculum schedule.
>
> Moreover, in the Linear–Gaussian case, we provide direct validation against the analytical optimum. The optimal design policy and stopping rule are available analytically, allowing us to compare learned behavior directly to ground truth. Figure 9 (in Appendix A.6.3) presents testing histograms showing that our learned policies accurately recover the analytical solutions for both design selection and stopping stages.
>
> **On baseline comparisons**: As discussed in our response to Weakness 2, the baseline methods cited (Huan & Marzouk, 2016; Foster et al., 2021) were not designed for problems with endogenous stopping and cannot be applied to our setting without fundamental architectural changes. There are no established baselines for optimal stopping in sequential information-theoretic BED.
>
> To nevertheless provide meaningful empirical validation, we pursue two complementary strategies:
> * **Comparison against analytical optimum (linear–Gaussian case)**: We demonstrate that our method recovers the known optimal solution, establishing correctness and convergence.
> * **Comparison against naïve stopping heuristics**: We evaluate against several intuitive alternatives that practitioners might adopt, including threshold-based rules (stop when accumulated reward reaches a threshold) and fixed-stage stopping rules. While straightforward to implement, these heuristics do not account for future value and cannot adapt to the evolving trade-off between information gain and experimentation cost. Our method achieves consistently higher expected return across all tasks, demonstrating both the necessity of a principled stopping rule and establishing the first viable baseline for this problem class.

---

### Author Response · Authors · 2025-11-23
**General Response**

We thank all reviewers for their thoughtful and detailed feedback. We appreciate the opportunity to clarify our contributions and address concerns about the scope and novelty of our work.

Our primary contributions are:
* **Theoretical framework**: We derive the optimal stopping theorem (Theorem 1) that characterizes when to stop experimentation based on continuation values, and extend the terminal–incremental equivalence (Theorem 2) to policy-dependent stopping settings, which are not previously established in the **information-theoretic BED** literature.

* **Algorithmic contribution**: We develop a practical policy-gradient algorithm (Theorem 3) that jointly learns design policies and optimal stopping rules within a unified MDP formulation, identify a circular dependency in joint optimization that destabilizes training, and resolve it via curriculum learning that gradually transitions from forced continuation to adaptive stopping.

* **Empirical validation**: We demonstrate that our method recovers analytical optimal solutions in tractable settings and outperforms naïve stopping heuristics in realistic, nontrivial problems.

In response to the reviewers' feedback, we have made the following revisions to strengthen the manuscript:
* **Expanded related work**: We have incorporated additional literature on stopping in adjacent domains (decision theory, Bayesian optimization, ranking and selection) to provide fuller context and more clearly distinguish our information-theoretic BED setting from related but distinct problem formulations.

* **Enhanced implementation guidance**: We have added discussion of the curriculum schedule's role, including comparison of different schedule choices (linear, exponential, sigmoid) and provided practical guidance and recommended parameter settings for practitioners.

* **Added computational considerations**: We have incorporated discussion of training and deployment costs, posterior approximation strategies for different problem scales, and the computational efficiency of learned policies at deployment time.

* **Improved clarity throughout**: We have corrected typographical errors, added explicit equation references in theorems, clarified terminology, and provided additional implementation details to improve readability and reproducibility.

We believe these revisions substantially strengthen the manuscript and address the reviewers' concerns. We respond to specific questions and weaknesses in detail below.

---

### Meta-Review · Area_Chair_ngrq · 2025-12-24

**Summary:**

This paper proposes a new reinforcement learning method that simultaneously optimizes designs and stopping for Bayesian experimental design (BED or BOED) tasks. The method goes beyond simple heuristics and is theoretically motivated and analysed. A policy gradient estimator is introduced, and a curriculum learning schedule is used to stabilise training. The method is demonstrated on a couple of simple tasks and is shown to outperform a standard baseline.

Strengths:
- The authors tackle an interesting and often neglected, but important, problem in an explicit and principled manner
- They give a theoretical analysis of the proposed method, elevating it to a principled stopping method rather than a heuristic
- Original take on casting problem as an explicit control process that incorporates design and stopping policies
- The paper is well written with good clarity

Weaknesses
- (PSxG, 7Jm2, VivY) The authors neglected to take into consideration highly relevant related stopping criteria presented in the Bayesian optimization (BO) literature (rectified), knowledge gradient (KG) policies, and stochastic processes (resolved)
- The authors propose a method that requires significant modification to existing BED methods for implementation of the stopping criterion.
- Limited applicability of evaluation and results, e.g. only simpler posterior design and constant cost cases are shown (PpyM, VivY)
- More ablations required of hand-designed stopping schedule and hyperparameter settings
- Wall clock/sample efficiency/computational complexity evaluation and discussion required (PpyM, 7Jm2, VivY, partially resolved)

The authors do an admirable job in addressing the misconceptions and methodological concerns of the reviewers. However, ultimately I believe the authors do not address the empirical evaluation concerns of the reviewers (experiments, ablations), and comparing this work to related work, the empirical evaluation does fall short (few experiments, only one run sample, no confidence intervals etc).

**Reviewer Concerns:**

Concerns addressed
- (PSxG, PpyM, 7Jm2) Incoporation of related literature from BO and similar fields.
- (PSxG) Highlight why incorporating a more simple additional “stop” action into existing EIG-based frameworks is not possible (e.g. non-differentiability) etc.
- (PpyM) clarified that they extend the theoretical results of DaD etc to the variable length horizon setting.
- (PpyM) clarified that the challenge addressed by the actor-critic is not posterior approximation, but estimation of the continuation value function.
- Non-Gaussian posteriors in experiments (source detection problem in experiment 5.2 is non-Gaussian)
- (PpyM) wall-clock results given (though for only one run), and added discussion of where computational bottlenecks arise (7Jm2).
- Clarification of how one-step knowledge gradient relates to time-horizon EIG methods that are the focus of this work. KG incorporated in to the related work section.
- Resolved multiple other small misconceptions from all reviewers

Concerns unaddressed
- Somewhat weak empirical evaluation: experiments perhaps somewhat too easy and too few (e.g. VivY: constant costs, lack of posterior calibration validation), and do not fully convince the reviewers of the efficacy of the stopping policy, despite theoretical analysis.
- Additional ablations and discussions of hyperparameters, hand designed schedules etc (PpyM, 7Jm2).
- Additional timing results with baselines, or clarification of how much computational effort is required over methods not estimating optimal stopping (all reviewers).

**Reviewer Scores:**

PSxG: 2 -> 4. Most concerns addressed, though it is hard to know if this reviewer would have been convinced to go past 4, as they seem to require convincing that this heavy modification is worth its complexity given the small performance gains in the experiments.

PpyM: 4 -> 6. Most concerns addressed by the authors, though no confidence intervals given on experiments/wall clock times.

7Jm2: 6 -> 6 or maybe 8. It appears as if the majority of this reviewers concerns have been addressed by the authors, with the exception of establishing the computational cost required to additionally estimate optimal stopping.

VivY: 4 -> 4 or 6. Many of VivY’s concerns addressed, but the criticisms of the experimental evaluation I believe remains unaddressed.

With all of this I believe the authors may have expected a score increase to either 5 or 5.5, however the empirical evaluation remains a weak point of the paper, and strong empirical evaluation is important for BOED (all reviewers has issues with this point).

---

### Decision · Program_Chairs · 2026-01-26

Reject